# DETECTION OF UNKNOWN UNKNOWNS IN AUTONOMOUS SYSTEMS

**Ayan Banerjee & Sandeep K.S. Gupta**
IMPACT Lab, School of Computing and Augmented Intelligence
Arizona State University, Tempe, Az
{abanerj3,sandeep.gupta}@asu.edu

## ABSTRACT

Unknown unknowns (U2s) are deployment-time scenarios absent from development/testing. Unlike conventional anomalies, U2s are not out-of-distribution (OOD); they stem from changes in underlying system dynamics without a distribution shift from normal data. Thus, existing multi-variate time series anomaly detection (MTAD) methods—which rely on distribution-shift cues—are ill-suited for U2 detection. Specifically: (i) we show most anomaly datasets exhibit distribution shift between normal and anomalous data and therefore are not representative of U2s; (ii) we introduce eight U2 benchmarks where training data contain OOD anomalies but no U2s, while test sets contain both OOD anomalies and U2s; (iii) we demonstrate that state-of-the-art (SOTA) MTAD results often depend on impractical enhancements: point adjustment (PA) (uses ground truth to flip false negatives to true positives, inflating precision) and threshold learning with data leakage (TL) (tuning thresholds on test data and labels); (iv) with PA+TL, even untrained deterministic methods can match or surpass MTAD baselines; (v) without PA/TL, existing MTAD methods degrade sharply on U2 benchmarks. Finally, we present sparse model identification–enhanced anomaly detection (SPIE-AD), a model-recovery-and-conformance, zero-shot MTAD approach that outperforms baselines on all eight U2 benchmarks and on six additional real-world MTAD datasets—without PA or TL. Code and data available in https://github.com/ImpactLabASU/U2Recognition.

## 1 INTRODUCTION

Autonomous systems such as unmanned aerial vehicles (UAV), autonomous cars (AC), and autonomous drug delivery (ADD) systems utilize complex amalgamation of interacting perception, decision making and actuation. Such complexity makes it practically infeasible to test for "all possible" operational scenarios. Test cases ignored during pre-deployment evaluation but that occur rarely during deployment, called "unknown unknowns" (U2), are a major cause of accidents (Maity et al., 2023; 2025). U2 detection is a significant problem with very few application specific solutions (Liu et al., 2020; Lakkaraju et al., 2017) in the image domain. In this paper, we present **SPIE-AD**, **SP**arse model **I**dentification **E**nhanced **A**nomaly **D**etection which detects U2 by continually mining the underlying dynamical model of variate inter-relationships and checking its conformance with the most likely model of normal operation.

The occurrence of U2 induces an effective violation of stationary property, as the underlying generating process experiences unmodeled changes that alter variate dependency structure over time (Fig. 1). U2s can potentially occur due to: a) **hardware failures**, which may not be monitored, e.g. mechanical failure in an aircraft resulting in an elevator getting stuck ($F8Stuck$) or moving slow ($F8Slow$), b) **unwanted software executions:** which may not immediately affect the input/output behaviour in anomalous ways, e.g. a change in the gravity parameter of a quadcoptor's altitude control software ($UAVSimG$), and c) **untested usage scenarios** manifested as external inputs to the system, which may not have a deviant measurement distribution parameter, e.g. an electromagnetic attack on a sensor decreasing its fidelity ($UAVEMA$) or a phantom meal, where an user of a insulin delivery ADD announces a meal without ingesting any to trick it for a high insulin dose. A natural question is that *can existing anomaly detection techniques be repurposed to detect U2s?*

The distinction between U2s and anomalies is subtle but critical. *While anomalies result in marginal distribution shift (Fig. 2 Panel A), U2s cause non-stationary changes in the dependency structure among variables, often without altering marginal distributions (Fig. 1 and 2 Panel B).* Hence to detect U2, we need *recovery and monitoring of underlying process model.*

State-of-the-art (SOTA) Multi-variate time series anomaly detection (MTAD) operate under the assumption of marginal distribution shift in other words out of distribution (OOD), and may use statistical regression methods e.g ARIMA (Schmidt et al., 2018), Kalman filter (Huang et al., 2023), principal component analysis based techniques (Shyu et al., 2003), autoencoders (Borghesi et al., 2019), long short term memory (LSTM) based deep learning (DL) techniques, transformers (Tuli et al., 2022) and most recently foundational models (Alnegheimish et al., 2024; Zhou et al., 2023). However, as conceded by recent research (Alnegheimish et al., 2024), existing MTAD techniques may fail to detect non-stationary changes in the underlying process. *As such, it remains to be seen whether existing MTAD pipelines can be used for U2 detection or not. We evaluate this question in this paper.*

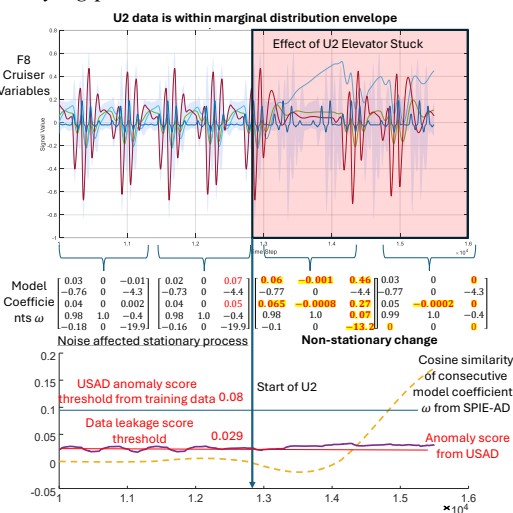

Figure 1: F8 elevator stuck U2 example. USAD (Audibert et al., 2020b) anomaly score threshold from training data does not detect U2. Threshold update using data leakage introduces significant false positives. Underlying model shows non-stationary change affected by U2.

The SOTA anomaly detection pipeline (Fig. 6 Panel A in Appendix) has three steps: a) **training**: that creates a high dimensional latent space representation of the normal operation using data that may or may not have anomalies but do not have anomaly labels, b) **validation**, that uses data with anomalies but without anomaly labels to learn a *anomaly score threshold* such that two fairly separated clusters are found in the validation set using the peaks over threshold method guided by the extreme value theory (Siffer et al., 2017), and c) **evaluation**, where anomaly score of successive windows of test data are computed and compared with the threshold to determine anomalous data. SOTA MTAD approaches have several assumptions that result in **unrealistic performance** and makes them unsuitable for U2 detection:

**A0: U2 detection problem does not conform to existing MTAD problem definitions -** SOTA MTAD problem can be of three types: i) **supervised MTAD**, where OOD anomaly data and labels available during training are used to develop models and classify the same anomalies seen during testing, ii) **unsupervised MTAD**, where training data with normal and OOD anomalous data without labels is used to learn the most likely model of normal operation and classify same types of anomalies in the testing phase, and iii) **semi-supervised MTAD**, where anomaly free training data is used to develop a model of normal operation and OOD anomalies are deviations from the normal. U2 detection problem assumes that training data with normal and OOD anomalous data without labels is available to learn the model of normal operation but test data consists of normal, OOD anomalous data and novel U2 scenario data with non-stationary dependency structures.

*Can we use existing few/zero-shot anomaly detection?* Recent LLM-guided few-shot anomaly detection techniques (Gao et al., 2024) are fundamentally unsuitable for U2 detection, as they require at least one prior U2 example. Existing zero-shot methods based on LLMs are limited to univariate time series (Alnegheimish et al., 2024) and thus do not extend to U2 settings. Zero-shot MTAD approaches convert time series into images and leverage vision–language models (VLMs) (He et al., 2025; Namura & Ichikawa, 2024). While promising, they assume the availability of signal-to-image pipelines and large-scale compute resources, which are often unavailable in resource-constrained autonomous and defense deployments. We therefore focus exclusively on U2 detection methods that operate directly on time-series signals without requiring image transformation.

**A1: Sensor data distribution shift due to anomaly:** For U2 scenario data there may not be a difference in the distribution parameters of the sensor outputs. Consider the example U2 scenario of wrongful Maneuvering characteristics augmentation system (MCAS) trigger in the fateful flight

of Lion Air (Curran et al., 2024). MCAS was designed to mask the flight characteristics changes that would have occurred on newer Boeing Max 8 aircrafts (Herkert et al., 2020). This implies that if MCAS is wrongfully triggered then by design it attempts to make the distribution parameters of the flight characteristics similar to a normal flight. Fig. 2 shows the data distribution of all sensors for anomalies and normal data in benchmark MTAD datasets in Panel A and for U2 and normal scenarios in Panel B. The Kolmogorov-Smirnov (KS) hypothesis test (KS, 2008) is used to compute the normalized maximum difference in cumulative distribution function (CDF) between normal and anomalous/U2 data (H = 1 implies the two distributions are statistically different with $(1 - P)$ probability. Higher value of the CDF difference implies more deviant distribution). While benchmark datasets exhibit distribution shift between anomalous and normal data, in U2 datasets, there is no statistically significant distribution shift between U2 and normal data.

**Technical difficulty in U2 detection violating A1:** A1's violation implies the raw sensor data may not have latent information to discriminate between normal and U2 classes. So, any data-driven feature based method e.g. existing MTAD methods may not be useful. While the sensor data distributions may not be discriminative, there maybe a change in functional relationship among the sensors. Panel C shows the underlying nonlinear dynamical model mined from U2 and normal data using SINDY-MPC (Kaiser et al., 2018) has significantly different distribution parameters. U2 detection could utilize modeling and monitoring of variations in such inter-relationships.

**A2: Use of data leakage to learn anomaly score threshold -** In SOTA MTAD techniques the validation set is often same as the test data (Appendix Section B). This leads to potential data leakage and overfitting of the model. It is standard machine learning practice to keep validation set separate from test data. By definition, no validation dataset with anomalies are available for U2 detection.

**Technical difficulty in U2 detection violating A2:** Violation of A2 entails zero shot U2 detection. To the best of our knowledge, there is only one solution for zero-shot MTAD (Audibert et al., 2020a). However, as identified by Kim et al. (2022), it has poor realistic performance. Solutions for univariate zero-shot anomaly detection including techniques with LLMs (Alnegheimish et al., 2024) are available, which, as admitted by the authors, are very difficult to

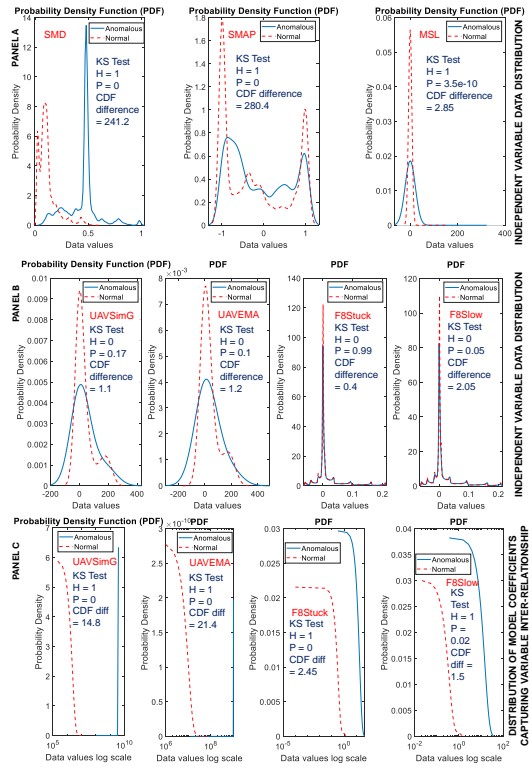

Figure 2: Panel A: Normal versus anomalous data distribution difference in benchmark datasets, (more evidence in supplement Table S3) Panel B: U2 datasets have negligible distribution difference with normal. Panel C: significant distribution difference in parameters of U2 versus normal data in the underlying sparse model space.

adapt to MTAD. The technical challenge is to detect anomalies with no knowledge about anomalous data distribution, which preempts any data driven discriminative feature learning. Note that a line of work, TimeseriesBench (Si et al., 2024), claims zero shot as the case where train and test data are disjoint. The definition of zero shot (Jayaraman & Grauman, 2014) used in this paper is different and requires identification of U2 from a description of its attributes without using any training data.

**A3: Unrealistic evaluation method-** According to Kim et al. (2022), the reported results in nearly all SOTA MTAD techniques have point adjustment (PA) (Su et al., 2019). This technique assumes anomalies to be contiguous segments, and it is sufficient for MTAD method to detect only one point in this segment as anomaly. The PA method inflates the precision by a significant amount (Wu & Keogh, 2023) in nearly all MTAD methods as seen in Fig. 7 in Appendix, which shows two most recent MTAD technique on benchmark datasets (SMAP, SMD, MSL) with/without PA (Liu et al., 2024). These results are also supported by Kim et al. (2022).

**Technical difficulty in U2 detection violating A3:** As highlighted in Kim et al. (2022), in many real-world datasets, anomalous or U2 events are often not abrupt and may result in an initial and final phase that have similar data distribution to normal data. So, if a MTAD method focuses only on purely data driven techniques for learning discriminative latent features, they may label the initial and final phases as normal. Without PA the performance may be unfairly under-reported.

**Main Technical Contribution:** *We present* **SPIE-AD***, that detects U2 by solving the general problem of zero-shot MTAD while violating the SOTA assumptions A1, A2 and A3.* The backbone of **SPIE-AD** are the *two fundamental theoretical contributions* of this paper: a) **robust sparse non-linear dynamical model recovery (MR)** from real-world multi-variate data using neural architectures with automated differentiation (AD) and b) **statistical conformance based model robustness interval extraction (CRIE)** method that can identify statistically relevant difference in recovered models. Utilizing these, *SPIE-AD* implements the following U2 detection pipeline (Panel B in Fig. 6 in Appendix): a) **training phase:** where **SPIE-AD** mines several models from training data snippets and defines a model robustness metric to quantify difference between models, b) **validation phase:** it uses part of the training data in the CRIE algorithm to determine a robustness metric interval for the most likely model of normal operation, and c) **evaluation phase:** it continually mines models from test data, computes robustness and compares with robustness interval to determine anomalies.

**Benchmark Contribution:** We introduce six synthetic benchmarks derived from U2 scenarios occurring in three different real-world systems including quadcopter, F8 cruiser, and automated insulin delivery (AID) and two novel real-world benchmarks from clinical study data. The hallmark of these benchmarks is that there is statistically insignificant distribution shift between the anomalous and normal data in each time series.

**Evaluation Contribution:** We first show that if we use PA (K = 0) and allow for data leakage in TL, then it is possible to develop an untrained simpleton machine (AnomalySimpleton in Fig. 7) that can beat SOTA MTAD techniques. While this was also argued in Kim et al. (2022), we propose a deterministic algorithm that gives consistent performance across benchmark datasets. We evaluate recently proposed MTAD techniques along with **SPIE-AD** under realistic scenarios where the precision is not augmented with PA (i.e. K = 100) and anomaly signatures in the form of validation set is not available for TL.

## 2 METHODOLOGY AND THEORETICAL FOUNDATIONS

We consider $n$ sensors each with time series $X^i$ for sensor $i$ forming a vector $X(t)$ over time where $t \in 0 \dots N/\mu$, where $\mu$ is the sampling frequency. The input / output time-series data from autonomous systems satisfies physical/chemical/mechanical/physiological properties of the real world system. Such properties are typically expressed using sparse non-linear dynamical systems:

$$\dot{X}(t) = f(X(t), \omega, t), \tag{1}$$

where $\omega$ is the set of $p$ model coefficients that defines the sparse model. An $n$-dimensional model with $M^{th}$ order non-linearity can utilize $\binom{M+n}{n}$ non-linear terms. A sparse model only includes a few non-linear terms $p << \binom{M+n}{n}$.

**U2 Definition:** Let $X(t)$ have a global marginal distribution $P(x)$. A window $W$ is a **U2 event** if: a) *Marginal distributions remain unchanged*: $P_t(x) \approx P(x), \forall t \in W$; but b) *Process becomes non-stationary*: $\omega(t) \neq \omega(t + \tau)$, for some $\tau$. Thus, a U2 event preserves marginal statistics but reflects a structural change in the system's underlying dynamics.

**Anomaly Definition:** A window $W$ is an **anomaly** if: a) *Window marginal drift occurs*: $P_t(x) \not\approx P(x)$, for some $t \in W$, as determined by a statistical test (Fig. 2); but b) *Underlying process remains stationary*: $\omega(t) = \omega(t + \tau), \forall t, \tau$. Thus, an anomaly induces local marginal deviation without altering the long-term distribution.

### 2.1 ROBUST SPARSE DYNAMICAL MODEL RECOVERY

Given $N$ time sequenced measurement of $X(t)$, sparse model recovery (SMR) aims to recover the coefficient $\omega$ such that the reconstructed measurements $Y(t)$ by solving the ordinary differential equation (ODE) in Eqn. 1 satisfies an error threshold $\epsilon$, i.e., $\sum_{t=1}^{N} ||Y(t) - X(t)||^2 < \epsilon$.

SMR is a well-researched problem with solutions ranging from L2 minimization techniques with sparse regression (SINDY-MPC) (Kaiser et al., 2018) to physics informed neural networks

(PINN) (Chen et al., 2021). It is generally acknowledged that SOTA MR techniques suffer significant performance degradation on data from real world systems (O'Brien et al., 2023). This implies that with low sampling frequency and high noise, the model coefficients $\omega_i$ and $\omega_j$ derived from two consecutive segments $[i, i + W]$, and $[j, j + W]$ of $X(t)$, with window size $W$ has significant variance. This is problematic for **SPIE-AD** since it will be difficult to distinguish between noise and real U2 scenarios and will hamper the false positives. **SPIE-AD** needs MR that is robust to measurement noise under low sampling rates.

To address robustness, **SPIE-AD** uses a novel neural network architecture with continuous time latent variable nodes, specifically liquid time constant neural networks (LTC-NN) as shown in Fig. 8 in Appendix (Hasani et al., 2021; Banerjee & Gupta, 2024b;a; Xu et al., 2026; 2025). Given a segment with $W$ samples, the SINDY-MPC technique is used to first recover a sparse model coefficient estimate $\omega(0)$. The data segment is passed through a fully connected network of $V$ LTC-NN cells in batches of $S_B$. The output of the LTC-NN nodes are fed to a dense linear layer with $\binom{M+n}{n}$ nodes with RELU activation function. The sparsity of $\omega(0)$, i.e. which elements of $\binom{M+n}{n}$ are "0" is used to dropout nodes of the dense layer. The output of the $i^{th}$ dense layer node are constrained within a range $[(1-\psi)\omega_i, (1+\psi)\omega_i]$, $\psi$ is a hyper-parameter. The weighted dense layer output is the refined estimate $\omega_{est}$ of the model coefficients and is fed to an ODE45 solver (Shampine et al., 2003) that reconstructs the signal $Y$. The loss is the mean square error between $X$ and $Y$ summed over dimensions and time steps. We show the effect of using this robust MR method on U2 detection. In Appendix Table S1, we show an ablation study with LTC-NN refinement removed on standard SMR benchmarks (Kaiser et al., 2018).

## 2.2 CONFORMAL INFERENCE FOR MODEL DEVIATION

Conformal inference (Tibshirani et al., 2019) is a distribution free method to identify whether a model, $\omega^v$, learned from validation data $[i, i + W]$ is in the distribution of the set of models $\Omega$ learned from training data. To compute model difference we use a robustness metric $\rho$ in Eqn. 2.

$$\rho(\omega^v, \Omega) = (\sum_{i=1}^{|\Omega|} \Omega_i^T \omega^v)/|\Omega|, \tag{2}$$

where $|\Omega|$ is the number of elements in the set $\Omega$ and $\Omega_i^T$ denotes transpose of an element in $\Omega$.

Let us consider that the training data has $k$ windows of size $W$ each, $X_1(1 \ldots W), X_2(1 \ldots W), \ldots X_k(1 \ldots W)$ where data is i.i.d in $\mathcal{R}^n \times \mathcal{R}^W$ drawn from a distribution $\mathcal{D}_X$. The SMR mechanism $L$ is used to derive coefficients $\omega_i \in \mathcal{R}^p$ from each $X_i$ such that reconstruction error is less than $\epsilon$. $L(.,.)$ is used to derive $\omega_{m+1}^v$ for $X_{m+1}, Y_{m+1}$ in validation data with no assumption on the $\mathcal{D}_{XY}$, hence no anomaly is required in validation set. Given the robustness function $\rho(.,.)$ in Eqn. 2, conformal inference creates a prediction band $C \subset \mathcal{R}^2$ based on $(X_1, Y_1), (X_2, Y_2), \ldots (X_m, Y_m)$ for a given *miscoverage level* $\alpha \in \{0, 1\}$, so that $P(\rho(\omega_{m+1}^v) \in C) \geq 1 - \alpha$. The prediction process can be encoded in Algorithm 1 CRIE (Appendix), which takes the i.i.d training data $(X_1, Y_1) \ldots (X_m, Y_m)$, miscoverage level $\alpha$ and the SMR method $L$ to provide the prediction interval.

The basic method is to divide the training set into two mutually exclusive subsets $I_T$ and $I_V$. The SMR method $L$ is used to derive $\omega_i$ for the segments $(X_i, Y_i) \in I_T$ and form the set $\Omega$. For each $\omega_i \in \Omega$, $\rho(\omega_i, \Omega_{/\omega_i})$ is computed, where $\Omega_{/\omega_i}$ denotes the set $\Omega$ with $\omega_i$ removed. Let $\sigma = avg_i(\rho(\omega_i, \Omega_{/\omega_i}))$ be the mean value of the robustness metric in the training set. From the validation set, $\omega_j^v$ is derived for $(X_j, Y_j) \in I_V$. The residual $\rho(\omega_j^v, \Omega) - \sigma$ is derived for every element in $I_V$, the residual is arranged in ascending order. The residual at the position $\lceil (|I_V|/2 + 1)(1 - \alpha) \rceil$ is used as the prediction range $d$. Theorem 2.1 in Lei et al. (2018) proves that the prediction interval at a new point $(X_{m+1}, Y_{m+1})$ is given by $L$ and satisfies Theorem 1.

**Theorem 1.** *If $\Omega$ is such that $||L(X_i, \omega_i) - X_i||^2 \leq \epsilon, \forall \omega_i \in \Omega$, for error margin $\epsilon$, then for a new $\omega_{m+1}^v$, $(X_{m+1}, Y_{m+1})$ Algorithm 1 (Appendix) ensures, $P(\rho(\omega_{m+1}^v, \Omega) \in [\sigma - d, \sigma + d]) \geq 1 - \alpha$.*

## 2.3 U2 DETECTION ALGORITHM

Utilizing Theorem 1 and the CRIE algorithm, we derived a robustness range that encodes the normal behavior without using the knowledge of U2. Our U2 detection mechanism in Algorithm 2 in Appendix takes windows of test data, uses the SMR technique to learn the model coefficients $\omega_i$, computes the robustness using Eqn. 2, and compares with the range obtained from CRIE.

## 2.4 SPIE-AD AND AUTOENCODER COMPARISON

**How SPIE-AD addresses A2?** The robust MR mechanism captures sensor inter-relationships unlike autoencoders that derive latent featurs of individual sensors. The **CRIE** algorithm learns a tight robustness range characterizing most likely normal operation in a distribution agnostic manner unlike autoencoders that need point estimation based EVT that assumes underlying distribution.

**How SPIE-AD addresses A3?** Unlike SOTA MTAD, **SPIE-AD** extracts low dimensional representation of the data which reduces entropy, making it easier to model normal scenarios. U2 scenario lead to exaggerated model deviation since the inter-relationship between variables become inconsistent. Hence, as seen in Table 3 & 4, **SPIE-AD** can achieve better overall precision without PA.

**How SPIE-AD addresses A1?** By learning an underlying model, SPIE-AD can exploit significant distribution differences in model space of U2 scenarios (Figure 2).

**Computational complexity:** Comprehensive analysis is provided in Section G in appendix.

## 3 RELATED WORK

Anomaly detection (AD) (Table 1) has a rich history starting from univariate AD with initial works employing Kalman Filter (Huang et al., 2023) and principle component analysis (PCA) (Shyu et al., 2003). PCA has been used for MTAD but not zero shot. The next generation MTAD techniques used statistical learn-

Table 1: Related works. Bold text − baselines. ¬ − assumption violation.

| Works | MTAD | Zero shot | ¬ A2 | ¬ A3 | ¬ A1 |
|---|---|---|---|---|---|
| Extended Kalman Filter (Huang et al., 2023) | No | Yes | Yes | Yes | No |
| Principle Component Analysis (Shyu et al., 2003) | Yes | No | No | No | No |
| **Time frequency anomaly detection (Zhang et al., 2022)** | Yes | No | No | No | No |
| **Frequency Interpolation Time Series (Xu et al., 2024)** | Yes | No | No | No | No |
| K nearest neighbor (Wang et al., 2020) | Yes | No | No | No | No |
| **Isolation Forest** (Liu et al., 2008) | Yes | No | No | No | No |
| **Light weight online anomaly detection** (Pevný, 2016) | Yes | No | No | No | No |
| **OmniANomaly** (Su et al., 2019) | Yes | No | No | No | No |
| **Anomaly transformers** (Xu et al., 2022) | Yes | No | No | No | No |
| **Graph attention networks** (Zhou et al., 2020) | Yes | No | No | No | No |
| **LSTM** (Hundman et al., 2018) | Yes | No | No | No | No |
| **Graph augmented normalized flows** (Zhao et al., 2022) | Yes | No | No | No | No |
| **One size fits all (Zhou et al., 2023)** | Yes | No | No | No | No |
| **Usupervised anomaly detection** (Audibert et al., 2020a) | Yes | Yes | Yes | No | No |
| CLIP zero shot image recognition (Pratt et al., 2023) | No | Yes | Yes | Yes | No |
| LLM Anomaly detection (Alnegheimish et al., 2024) | No | Yes | Yes | Yes | No |
| **SPIE-AD** | **Yes** | **Yes** | **Yes** | **Yes** | **Yes** |

ing methods such as K nearest neighbors (Wang et al., 2020) or Isolation Forest (iForest) (Liu et al., 2008) mechanisms or light weight online anomaly detector (LODA) (Pevný, 2016). Such techniques are not tested for zero shot MTAD and also had poorer overall performance on real world data (Liu et al., 2024). Time series analysis methods have also been used for MTAD such as time frequency domain approaches (Zhang et al., 2022) or frequency interpolation methods (Xu et al., 2024). The current generation of MTAD techniques uses DL such as LSTM (Hundman et al., 2018), variational autoencoders (OmniAnomaly) (Su et al., 2019), anomaly transformers (AT) (Xu et al., 2022), graph augmented normalized flows (GNAF) (Zhao et al., 2022), and Graph Attention Networks (GAT) (Zhou et al., 2020) or even foundational models such as one size fits all (OFA) approach (Zhou et al., 2023). These MTAD techniques however use the workflow described in Fig. 6 and do not achieve zero shot MTAD. While U2 has been explored in the image domain using large vision models such as CLIP (Pratt et al., 2023) such methods are not directly applicable to MTAD. We are aware of two works, i) unsupervised anomaly detection (USAD) that performs zero shot MTAD (Audibert et al., 2020a) using autoencoders, and ii) and use of large language models (LLMs) to perform U2 in univariate timeseries (Alnegheimish et al., 2024). The USAD technique still reports anomaly detection accuracy with PA (A3) and TL (A2), and relies on difference in distribution shift between normal and anomalous class (A1).

## 4 EVALUATION

We perform three types of evaluation: **A)** effects of using test set as validation set (A2) and PA (A3) on MTAD performance. We show that an untrained statistical method can beat SOTA learning based systems with A2 and A3. **B)** performance comparison of SPIE-AD and SOTA baselines under violation of A2 and A3 on U2 benchmarks that have no distribution shift between anomaly and normal data (violates A1). **C)** performance comparison of SPIE-AD and SOTA baselines on real world univariate and multivariate datasets. Using the large univariate UCR dataset we perform statistically robust evaluation of sensitivity of SPIE-AD on window size $W$ in appendix Section F.

**AnomalySimpleton:** We propose an untrained deterministic thresholding algorithm that exploits PA and test data distribution i.e. data leakage to provide anomaly detection performance on par with state-of-the-art learning techniques. In this method, a specific window $W$ of data is selected from the train data. Statistical properties of the train data window $W$ such as mean $\psi_{train}$, standard deviation $\sigma_{train}$, and skewness $\kappa_{train}$ is computed. For each test data window of length $W$, the same statistics are computed. If the deviation of the test statistics is more than P% of the train statistics, then the test data window is classified as anomalous else it is not anomalous. The window $W$ and the test statistics $P$ is used to obtain two maximally separated clusters in the test data. This is done through brute force search over several $W$ and $P$ options. For each benchmark real world data this window and threshold seach is performed from scratch.

**Benchmarks:** We used 14 datasets (Table 2) to evaluate **SPIE-AD**, out of which 6 are synthetic and 2 real world U2 datasets, while 3 are real world MTAD datasets and another 3 are popular large scale univariate real world datasets taken from TimeSeriesBench (Si et al., 2024).

We utilize both synthetic and real world U2 datasets. Detailed description of synthetic U2 datasets are provided in the appendix while real world data is described below. While the synthetics datasets highlights the efficacy of **SPIE-AD** in U2 detection while violating $A2$, $A3$, and $A1$, the real world anomaly datasets show the generality of **SPIE-AD**.

Table 2: Datasets. Train, T, Test, Te, Real world, R, Synthetic, S

| Dataset | Dim | Samples (T/Te) | Anomaly / U2 % | Type |
|---|---|---|---|---|
| Electromagnetic attack (UAVEMA) | 3 | 240K/242K | 29.75% | S |
| Simulated g change (UAVSimG) | 3 | 240K/274K | 11.7% | S |
| F8 cruiser stuck elevator (F8Stuck) | 4 | 877K/237K | 9.2% | S |
| F8 cruiser slow elevator (F8Slow) | 4 | 877K/843 K | 1.4% | S |
| AID phantom meal (AIDPhantom) | 4 | 260K/240K | 12% | S |
| AID cartridge error (AIDCartridge) | 4 | 260K/302K | 11.5% | S |
| Medtronic Cartridge error | 3 | 256K/518 K | 3% | R |
| EEG Seizure data | 15 | 512K/675 K | 7% | R |
| Server Machine Dataset (SMD) | 38 | 708K/708K | 4.16% | R |
| Soil Moisture (SMAP) | 25 | 135K/427K | 13.13% | R |
| Mars Science Lab Rover (MSL) | 55 | 58K/73K | 10.7% | R |
| 250 UCR anomaly dataset | 1 | 5M/13M | 0.4% | R |
| Yahoo dataset | 1 | 400K/780K | 12% | R |
| NAB dataset | 1 | 54K/104K | 6% | R |

**Real world datasets:** A) Benchmark MTAD datasets available in Su et al. (2019), and large univariate datasets UCR, Yahoo and NAB database available in Wu & Keogh (2022); Si et al. (2024).

Table 3: Comparison of **SPIE-AD** against baselines for synthetic U2 benchmarks (S = synthetic). **SPIE-ADS** uses SINDY-MPC, **SPIE-ADL** uses LTC-NN. $^+$ denotes with point adjustment (PA).

| Approach | F8Stuck S | | | F8Slow S | | | UAVSimG S | | | UAVEMA S | | | AIDPhantom S | | | AIDCartridge S | | |
|---|---|---|---|---|---|---|---|---|---|---|---|---|---|---|---|---|---|---|
| | Pr | Re | F1 | Pr | Re | F1 | Pr | Re | F1 | Pr | Re | F1 | Pr | Re | F1 | Pr | Re | F1 |
| Omni$^+$ | 91.2 | 72.7 | 80.9 | 88.4 | 71.1 | 78.8 | 92 | 77.1 | 83.9 | 90 | 67.3 | 77.0 | 94 | 76.1 | 84.1 | 97 | 59.7 | 74 |
| Omni | 41 | 26.8 | 32.4 | 65 | 28.1 | 39.2 | 32 | 19.7 | 24.4 | 29 | 16.8 | 21.3 | 19.1 | 16.5 | 17.7 | 65 | 31.9 | 43 |
| AT$^+$ | 100 | 78.6 | 88 | 100 | 58.7 | 74.1 | 100 | 59.2 | 74.2 | 90 | 56.1 | 69.1 | 91 | 56.3 | 69.7 | 100 | 59.2 | 74 |
| AT | 85.5 | 75.8 | 80.3 | 34.2 | 32.8 | 33.5 | 35 | 33.5 | 34.2 | 33.9 | 32.4 | 33 | 34 | 32 | 33 | 34.3 | 33.8 | 34 |
| iForest$^+$ | 100 | 78.6 | 88 | 100 | 47.5 | 64.4 | 100 | 50.8 | 67.6 | 88.5 | 46.2 | 60.7 | 98.6 | 45.9 | 62.6 | 91.2 | 42.1 | 57.6 |
| iForest | 14 | 33 | 19.6 | 9.8 | 8.2 | 8.9 | 10.6 | 8.5 | 9.4 | 8.6 | 7.6 | 8.1 | 9.5 | 8.1 | 8.7 | 9.5 | 7.9 | 8.6 |
| LODA$^+$ | 100 | 72.6 | 84 | 100 | 20.7 | 34.3 | 96.9 | 18.5 | 31 | 88.5 | 14.9 | 25.5 | 95.8 | 16.8 | 28.6 | 99.2 | 17.2 | 29.4 |
| LODA | 88 | 70 | 78 | 60.7 | 13.7 | 22.4 | 50.7 | 11 | 18 | 35 | 8.6 | 13.8 | 35.8 | 9.4 | 14.9 | 36.4 | 9.7 | 15.3 |
| LSTM$^+$ | 100 | 88 | 93 | 100 | 47.8 | 64.7 | 91.8 | 20.2 | 33.2 | 100 | 21.2 | 35 | 99.9 | 20.3 | 33.8 | 96 | 18.6 | 31 |
| LSTM | 77 | 85 | 80 | 61 | 35.8 | 45.2 | 59.4 | 13.2 | 21.6 | 60.8 | 14.2 | 23 | 58.6 | 12.6 | 20.7 | 54.7 | 12.1 | 19.9 |
| USAD$^+$ | 100 | 72.1 | 83.8 | 100 | 23 | 37.4 | 92.6 | 21.8 | 35.3 | 90.3 | 21.6 | 34.9 | 94.6 | 25.2 | 39.8 | 97.1 | 28.6 | 44 |
| USAD | 81 | 67.7 | 74 | 55.3 | 14.2 | 22.6 | 51.2 | 12.3 | 19.8 | 49.2 | 12.1 | 19.4 | 52.6 | 12.1 | 19.7 | 58 | 8.8 | 15.2 |
| GANF$^+$ | 100 | 86 | 92.5 | 100 | 58 | 73 | 100 | 92.2 | 96 | 100 | 97 | 98.5 | 96.7 | 61.5 | 75 | 92.8 | 56.1 | 70 |
| GANF | 61 | 79 | 68.8 | 3.2 | 4.3 | 3.7 | 51.4 | 85 | 64.3 | 0.9 | 24.7 | 1.8 | 3.2 | 4.5 | 3.8 | 2.1 | 2.7 | 2.4 |
| GAT$^+$ | 100 | 85.2 | 92 | 100 | 47.2 | 64.1 | 99.2 | 48.3 | 65 | 86.4 | 44.6 | 58.8 | 92.8 | 48.1 | 63.4 | 99 | 49 | 65.6 |
| GAT | 71.4 | 80.5 | 75.7 | 58.9 | 34.5 | 43.5 | 59.2 | 32.3 | 41.8 | 50.4 | 28 | 36 | 54.5 | 28.9 | 37.8 | 57.2 | 30.3 | 39.7 |
| OFA$^+$ | 82.1 | 87.5 | 84.7 | 65.9 | 43.2 | 52.2 | 66.2 | 72.3 | 69.1 | 70.4 | 68 | 69.2 | 74.5 | 77.1 | 75.8 | 81.3 | 87.4 | 84.2 |
| OFA | 21.4 | 4.5 | 7.4 | 21.9 | 9.7 | 13.4 | 37.5 | 22.1 | 27.2 | 20.3 | 8.5 | 12 | 31.3 | 18.3 | 23.1 | 21.7 | 10.1 | 13.8 |
| FITS$^+$ | 91.4 | 70.5 | 79.6 | 81.3 | 74.2 | 77.6 | 81.9 | 82.3 | 82.1 | 80.1 | 76 | 78 | 74.3 | 88.1 | 80.6 | 97.2 | 70.1 | 81.5 |
| FITS | 21.4 | 8.6 | 12.3 | 48.1 | 14.3 | 22.05 | 17.3 | 21.9 | 19.3 | 80.4 | 2.4 | 4.7 | 24.5 | 18.4 | 21.0 | 14.7 | 40.1 | 21.5 |
| TFAD$^+$ | 82.1 | 77.4 | 79.7 | 78.2 | 84.3 | 81.1 | 91.9 | 82.3 | 86.8 | 80.4 | 88 | 84.0 | 71.5 | 78.9 | 75.0 | 87.2 | 80.3 | 83.6 |
| TFAD | 11.2 | 30.4 | 16.4 | 9.8 | 21.7 | 13.5 | 29.5 | 12.4 | 17.5 | 21.9 | 8.7 | 12.4 | 14.7 | 31.8 | 19.9 | 17.7 | 21.4 | 19.4 |
| **SPIE-ADS$^+$** | 87.3 | 100 | 93.2 | 54.8 | 100 | 71 | 82 | 100 | 90.1 | 91.1 | 100 | 95.4 | 94 | 98.1 | 96 | 95.3 | 93 | 94.1 |
| SPIE-ADS | 86.7 | 94.5 | **90.4** | 51 | 85 | **66** | 82 | 99.9 | 90.1 | 91.1 | 100 | **95.4** | 91 | 96 | **93.4** | 92 | 85 | **88.4** |
| **SPIE-ADL$^+$** | 88.9 | 100 | 94 | 55.1 | 100 | 73 | 91 | 100 | 95.3 | 93.2 | 100 | 96.5 | 94.1 | 99 | 96 | 95 | 94 | 94.1 |
| **SPIE-ADL** | 88.7 | 95.1 | **92** | 58 | 93 | **70** | 89 | 99.9 | **94.2** | 93.2 | 100 | **96.5** | 92.1 | 99 | **95.4** | 91 | 92 | **91.5** |

B) Real world U2 data, for cartridge occlusion in Medtronic 670 G obtained from JAEB center (JAEB center, 2023) and clinical electroencephalography (EEG) data capturing sudden onset of epileptic seizure (Ghorbanian et al., 2015).

**Baseline Techniques:** We compare **SPIE-AD** with a combination of time series, deep learning, autoencoder, and foundational model based techniques highlighted in italics in Table 1. In addition,

we also compare with some table topper univariate AD methods reported in Wu & Keogh (2023); Lee et al. (2024b); Si et al. (2024) (Table 6 and 7 in Appendix).

**SPIE-AD implementation:** We implemented two variations of **SPIE-AD**: a) **SPIE-ADS**, where the model recovery part is solely SINDY-MPC, and b) **SPIE-ADL**, where the model recovery part is SINDY-MPC augmented with the LTC-NN neural architecture with AD. For the SINDY-MPC implementation we used the code from Kaiser et al. (2018). For the LTC-NN neural architecture, we updated the base code available in Hasani (2024). The **CRIE** and **U2 detection** algorithms were implemented using Matlab 2022b.

*Hyper-parameter optimization:* As highlighted in Fig. 6, there is a hyper-parameter optimization step in **SPIE-AD** during the training process. The hyper-parameters include: a) miscoverage level $\alpha$ that determines the robustness interval width $d$, b) the polynomial order of SMR technique, c) the sparsity level of the model, and the window size $k$. These parameters were determined only using the training data with the objective to include atleast $r > 80\%$ points of the training dataset within the robustness interval while minimizing $d$. The hyper-parameter optimization approach was brute-force and performed for each application (Section B.4).

**Baseline Implementation:** We used the MTAD tools and pipeline established in Liu et al. (2024) for baseline implementations. In all baseline implementations except USAD, we observed that removing labels from validation set reduced the precision and recall to near zero. Indicating that a pure zero-shot MTAD implementation with baselines is not possible without significantly altering the methods. Hence, in our comparison all baselines were non zero-shot MTAD except for USAD and **SPIE-AD**. For all implemented techniques we show two cases with and without PA.

**Evaluation metrics:** We use standard metrics: Precision (Pr), Recall (Re), and F1 score (Liu et al., 2024). For the univariate

Table 4: Comparison of **SPIE-AD** against baselines for real-world U2 benchmarks (R = real world).

| Approach | Medtronic R | | | Epilepsy R | | |
|---|---|---|---|---|---|---|
| | Pr | Re | F1 | Pr | Re | F1 |
| Omni$^+$ | 41.1 | 62.3 | 49.5 | 25 | 17 | 20.2 |
| Omni | 2.1 | 5.2 | 3 | 5.1 | 14.2 | 7.5 |
| AT$^+$ | 51.3 | 72 | 59.9 | 43.4 | 61 | 50.7 |
| AT | 30 | 20 | 24 | 40 | 50 | 44.4 |
| iForest$^+$ | 61.3 | 54 | 57.4 | 66.4 | 71.2 | 68.7 |
| iForest | 23.1 | 31.2 | 26.5 | 33.1 | 33.1 | 33.1 |
| LODA$^+$ | 45.1 | 65.3 | 53.3 | 53 | 55 | 54 |
| LODA | 3.4 | 91 | 6.6 | 4.1 | 15.4 | 6.5 |
| LSTM$^+$ | 69 | 60 | 64.2 | 15.1 | 18 | 16.4 |
| LSTM | 2.7 | 10.5 | 4.3 | 4.4 | 21 | 7.3 |
| USAD$^+$ | 67 | 71 | 68.9 | 25 | 74 | 37.4 |
| USAD | 31 | 43 | 36 | 12 | 65 | 20.2 |
| GANF$^+$ | 43 | 84 | 56.9 | 63 | 75 | 68.5 |
| GANF | 4.1 | 91 | 7.8 | 33 | 35 | 34 |
| GAT$^+$ | 64 | 60 | 61.9 | 34 | 34 | 34 |
| GAT | 13.2 | 29.1 | 18.2 | 12 | 65 | 20.2 |
| OFA$^+$ | 69 | 71 | 70 | 43 | 57 | 49 |
| OFA | 60 | 56 | 57.9 | 39 | 55 | 45.6 |
| FITS$^+$ | 65 | 40 | 49.5 | 24 | 35 | 28.5 |
| FITS | 55 | 37.5 | 44.6 | 10.1 | 17.5 | 12.8 |
| TFAD$^+$ | 65.2 | 60 | 62.5 | 43 | 30 | 35.3 |
| TFAD | 21.5 | 15 | 17.7 | 17.5 | 19 | 18.2 |
| **SPIE-ADS**$^+$ | 69 | 71 | 70 | 64 | 79 | 70.7 |
| **SPIE-ADS** | 67 | 70 | **68.5** | 60 | 77 | **67.4** |
| **SPIE-ADL**$^+$ | 69 | 72 | 70.5 | 65 | 79 | 71.3 |
| **SPIE-ADL** | 69 | 71 | **70** | 64 | 75 | **69** |

real-world UCR database, the event-based AD accuracy is used as in Timeseriesbench (Si et al., 2024). If the detected anomaly sample is in $\pm$ 100 samples of the anomaly start point, accuracy is 1, else 0. Plus we show execution times of all methods for real world datasets. For MTAD methods that depend on TL, a threshold independent metric, volume under the surface (VUS) of the area under the precision recall curve (AUC-PR) is reported Paparrizos et al. (2022). SPIE-AD is not dependent on an anomaly threshold. One way to incorporate this is to compute VUS by changing the coverage level $\alpha$. We report VUS and AUC-PR in appendix Table S4.

## 5 RESULTS

We first show the inefficacy of the evaluation strategy used in state of the art MTAD techniques. We then evaluate the performance of **SPIE-AD** and compare with baseline on U2 benchmarks. We then compare **SPIE-AD** performance on real datasets. Here we also perform two ablation studies: a) removing point adjustment, and b) removing acess to validation datasets with anomalies. Our lessons learned from AnomalySimpleton experiment is available in Section D in Appendix.

### 5.1 U2 DETECTION PERFORMANCE EVALUATION

Table 3 and 4 show that **SPIE-ADS** outperforms SOTA on the F1 score for the case without PA - implying it has better precision and recall and does not need PA. Methods such as anomaly transformers (AT) do outperform **SPIE-AD** in F1 metric with PA - implying **SPIE-AD** does miss some legitimate events as evidenced by the slightly higher recall.

Interestingly, among the DL methods, AT has the highest difference between F1 scores with and without PA. However, AT has the highest F1 score for $F8Slow$. This entails that while anomaly trasnformer is good at detecting U2, albeit very late. Further, SPIE-AD also outperforms the only other zero-shot MTAD methods USAD. USAD also has a significant difference in metrics with/without PA (A3). SPIE-AD requires no such assumptions.

For nearly all cases **SPIE-ADL** consistently outperforms **SPIE-ADS**, showing the robustness improvement property of the LTC-NN approach in Fig. 8 in Appendix. However, the difference is much lower and given that LTC-NN architecture is much more complex than SINDY-MPC, one may wonder why it's necessary. A point is that all these benchmarks are synthetic; hence are much less noisy reducing its need. The need for LTC-NN is illustrated in real data.

## 5.2 REAL WORLD ANOMALY DETECTION PERFORMANCE

**Multi-variate:** Table 6 in supplement shows the performance of SPIE-AD on real datasets and compares it to recent DL based MTADs and unsupervised methods. In real data, SPIE-AD outperforms SOTA on F1 score without PA. On real data, we see the largest benefit of using the LTC-NN.

**Univariate:** Maximum event-wise AD accuracy of SPIE-ADS was 75.6% on UCR database. Compared to the leaderboard in Lee et al. (2024a), SPIE-ADS beats the SOTA by 4.8%.

**Ablation Studies:** For each real dataset we created three configurations: with point adjustment and validation set (PA + V), without PA (¬ PA), and without validation set i.e. zero shot (¬ V). It is observed that as expected the F1 score of SOTA DL techniques reduce drastically without PA. The USAD has lesser effect, while the SPIE-AD methods have the least ef-

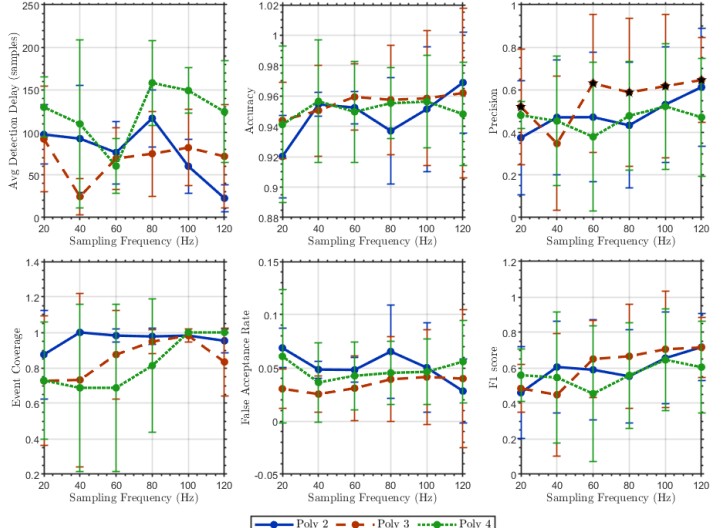

Figure 3: Performance with respect to sampling frequency, and library size (higher polynomial order results in combinatorially larger library) Low noise case. High noise case in Appendix Fig. 10.

fect of PA. Moreover, removal of validation set reduces the F1 score to near zero for anomaly transformer and GNAF approaches. Both USAD and SPIE-AD have higher F1 score for zero-shot MTAD, with SPIE-AD outperforming USAD.

## 5.3 SPIE-AD ANALYSIS

In this section, we perform the following analysis: *a) Evaluate the sensitivity of SPIE-AD U2 detection performance to library size, sampling rate, and noise:* For this experiment, we consider all the U2 benchmarks synthetic and real world combined

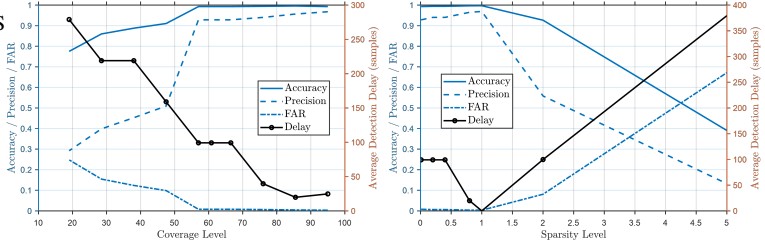

Figure 4: Effect of changing coverage level $1 - \alpha$ and sparstiy of SINDY-MPC on SPIE-AD performance.

and report averaged results. We resampled all the mutlivariate data to 100 Hz and then varied the sampling rate from 20 Hz to 120 Hz in steps of 20 Hz. For each frequency level, we varied the maximum polynomial order from 2 to 4 in steps of 1. We considered two noise conditions: i) 20 dB signal to noise ratio (SNR), achieved by adding Gaussian noise to each signal dimension, this represents low noise scenario, and ii) high noise scenario with 5dB SNR. In Figures 3 and 10 (in Appendix) we report additional performance metrics: 1) false acceptance rate (FAR), 2) event coverage, 3) time delay in terms of samples to detect U2, 4) accuracy, 5) precision, and 6) F1 scores. The figures show that as sampling frequency reduces all performance metrics become poorer, with polynomial order 3 providing the best overall results for SPIE-ADL. Moreover as noise increases accuracy, precision, FAR, F1 reduce but time delay has more variance and sometimes is better for higher noise scenario. This happens when noisy data immediately precedes U2 event and SPIE-AD classifies noise as U2, but the detection delay results in false U2 detection right after actual U2 occurrence.

*b) Evaluate the effect of different SINDY backbones on SPIE-AD, U2 detection performance:* There are several SINDY variants as summarized in Table 11 in Appendix. In Fig. 11 in Appendix we compare the effect of using W-SINDY specifically designed for high noise scenarios, and SINDY without control on SPIE-AD. It shows that SINDY-MPC is the best SINDY variant in terms of all performance metrics. W-SINDY and SINDY without control suffers because they do not handle exogenous inputs. We also observed the effect of adding a non-polynomial (sine) term in the library, which resulted in significant drop in performance metrics. This heavily depends on the stability of the STRIDGE algorithm in evaluating regression on non-polynomial functions.

*c) Evaluate the effect of different continuous depth neural networks on execution speed and U2 detection performance:* SINDY variants are the fastest as shown in columns 4, 5, and 6 on real world data in Table 6 in supplement. We further valuated the use of continuous time recurrent neural networks (CT-RNN) and NODE replacements of LTC-NN. These variants improved execution time ($1.2 \times$ for NODE, and $1.7 \times$ for CT-RNN) but resulted in poorer performance (Fig. 12 in Appendix).

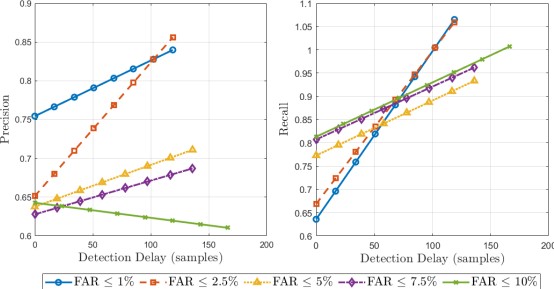

*d) Evaluate the effect of miscoverage level, and sparsity settings on U2 detection performance:* We varied the coverage level $1 - \alpha$ (or miscoverage level $\alpha$) from 0.95 to 0.2 for SPIE-ADL and performed experiments for the F8 example (most complex model). In SINDY-MPC the sparsity threshold controls sparsity such that if values are less than the threshold they are ignored and the representation becomes sparser. We introduce a sparsity level from 0.001 to 5.0, which is a multiplicative factor to this threshold. As sparsity level increases the underlying model becomes sparser. Fig. 4 shows that SPIE-AD is sensitive to both miscoverage level

Figure 5: Precision / Recall v.s. detection delay for fixed FAR budgets (trend lines only).

and sparsity. As miscoverage level increases all performance metrics monotonically become worse. However, sparsity dependency is more interesting. From Fig. 4, we see that there is an optimal sparsity level for which the model performs best. Additional experiments show that this optimal sparsity level varies for different underlying systems.

*e) Evaluating the trade-off between Precision, Recall and U2 detection latency under fixed false acceptance rate budget:* In this experiment, across all the U2 synthetic and real world datasets, we fixed the average FAR to 0.01, 0.025, 0.05, and 0.1 and plotted the precision / recall against detection delay. Fig. 5 shows that if for a fixed FAR budget, when the delay in detection increases both precision and recall increase. This means that larger number of samples result in better U2 detection accuracy. However, this is not true if we allow for higher FAR. At FAR budget of 10%, the precision decreases with increased delay. SPIE-AD performance is poorer with smaller U2 events. If U2 length is small then the delay increases and SPIE-AD identifies data points after the U2 event as U2 and hence results in higher false positives reducing precision.

## 6 CONCLUSIONS

In this paper, we introduced **SPIE-AD** a methodology for identifying 'unknown-unknown' (U2) errors in AI-enabled autonomous systems. U2 can arise due to unpredictable human interactions and complex real-world usage scenarios, potentially leading to critical safety incidents through unsafe shifts in the distribution of the inter-relationships among the variables in operational data. SPIE-AD performs zero shot anomaly detection and hence does not require signature of the U2 scenario or detection. Validation across diverse contexts such as zero-day vulnerabilities in unmanned aerial vehicles, hardware failures in autonomous insulin delivery systems, and design deficiencies in aircraft pitch control systems such as Maneuvering Characteristics Augmentation Systems (MCAS), demonstrates our framework's efficacy in preempting unsafe data distribution shifts due to unknown-unknowns. This methodology not only advances unknown-unknown error detection in AAS but also sets a new benchmark for integrating physics-guided models and machine learning to ensure system safety. Mining the underlying model of a dynamical system has several applications including detection of stealth cheating scenarios in AI systems much like the Volkswagon emission cheating case, or also biometric liveness detection. We have not only shown efficacy of SPIE-AD on U2 datasets but also demonstrated its generality in detecting any anomalous scenarios through the usage of standard real world datasets.

## 7 ACKNOWLEDGMENTS

This work was partially funded by DARPA (AMP, N6600120C4020; FIRE, P000050426), the NSF (FDT-Biotech, 2436801), NIH R21 grant (1R21HL175632) and the Helmsley Charitable Trust (2-SRA-2017-503-M-B)

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

## 8 REPRODUCIBILITY

We will make our dataset public through the MTAD tools and techniques github page (Liu et al., 2024) for the general research community to develop novel U2 detection schemes. We have also shared our code in Abstract.

## 9 ETHICAL CONSIDERATIONS

One of the components of SPIE-AD is recovering underlying model. One of the applications of SPIE-AD is digital twins. An unethical usage is impersonation. Thus, careful ethical evaluation is required when integrating such systems in medical practice. Another issue is that SPIE-AD is only a U2 detection mechanism. In its current form it cannot be used to explain the reasons behind the U2 occurrence. Such black box models can become problematic if false positives lead to usage of critical intervention. Hence proper safeguards should be placed to vet the U2 decisions from SPIE-AD.

## 10 LIMITATIONS

SPIE-AD faces challenges in determining point anomalies that last very few samples. In the SMD SMAP and MSL datasets, anomalies that last $< 5$ samples are missed consistently. Moreover, as seen in Fig. 9 SPIE-ADS performance is sensitive to the window size chosen for the CRIE algorithm. Hence, an important future work is to formally evaluate the sensitivity of SPIE-AD to window length.

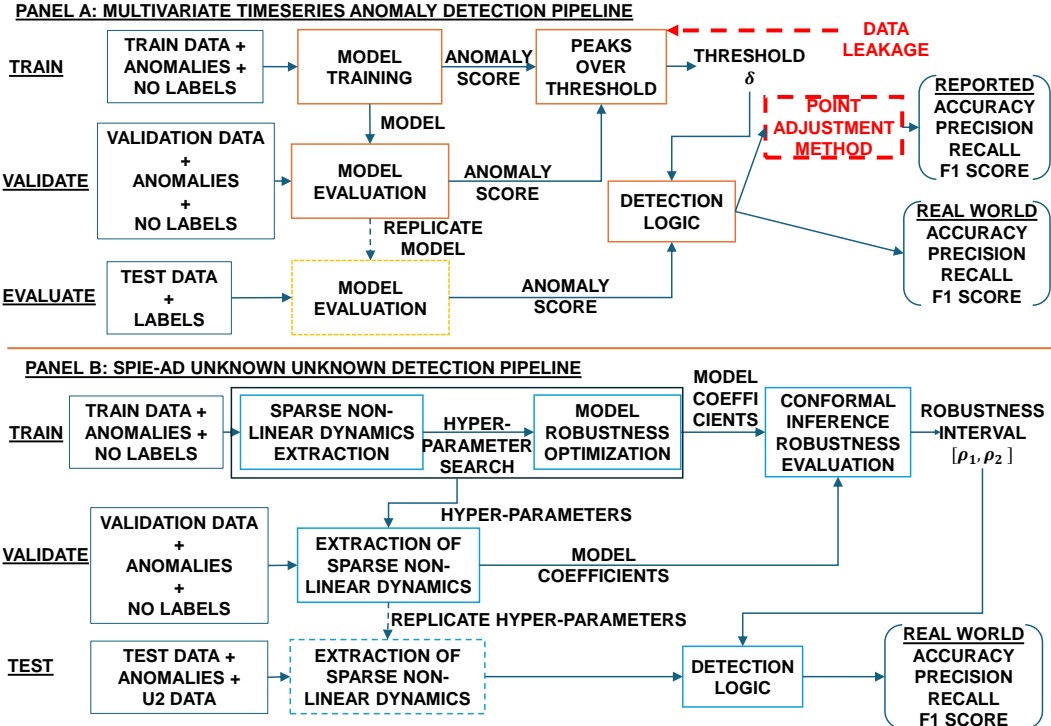

Figure 6: Panel A: SOTA MTAD pipeline with the identified issues highlighted by dashed arrows and boxes. Panel B: SPIE-AD's approach for solving zero-shot MTAD problem.

## A  ALGORITHMS CRIE AND U2DETECT

---

**Algorithm 1 CRIE($\{X_i\}_{i=1}^{N}$, $\alpha$, $\rho(.,.)$, $L$)**

---

1: **input** Data $\{X_i\}_{i=1}^{N}$, miscoverage level $\alpha$, robustness function $\rho$, SMR $L$
2: **output** Prediction range $d$
3: Split $\{1, \dots, N\}$ into two equal sized subsets $I_T$ and $I_V$.
4: $\omega_i = L((X_i) : i \in I_T)$
5: $\omega_j^v = L((X_j) : j \in I_V)$
6: Average robustness $\sigma = avg(\rho(\omega_i, \Omega_{/\omega_i}))$
7: For each $\omega_j^v$ compute residual $R_j = \rho(\omega_j^v, \Omega) - \sigma$
8: **return** $d$ = the kth smallest value in $\{R_j : j \in I_V\}$, where $k = \lceil (|I_V|/2 + 1)(1 - \alpha) \rceil$

---

## B  EVIDENCE OF DATA LEAKAGE

(refer to line 196 to 200 in the `data_loader.py` code in `https://github.com/thuml/Anomaly-Transformer`). Also it is evident from the code available in the github link for TSB-AD Liu & Paparrizos (2024). The fit function in line 97 in TSB-AD/TSB_AD/models/USAD.py extracts train and validation data and and function decision_function (line 180) extracts test data using dataloader. TSB-AD/TSB_AD/main.py line 39 that loads the data. For supervised methods in main.py line 39 instantiates *data* and line 44 extracts *data_train* from which train and validation data are extracted. However, at line 50 data_train is passed as train data *but the whole data is passed as test data*.

**Snippet of Results** (AT – Anomaly Transformer, GNAF – Graph Augmented Normalizing Flows)

| SPIE-AD evaluation on benchmarks SMD, SMAP, MSL [Xu et al. ICLR' 22] | Average F1 scores (F1) and precision (P) across three benchmark datasets for MTAD (Exhaustive metrics in Results Section) | | | |
|---|---|---|---|---|
| | Validation with data leak | | Validation without data leak | |
| | With point adjustment | Without point adjustment | With point adjustment | Without point adjustment |
| AT [ICLR'22] | F1:90 $\pm$2,P:98 $\pm$3 | F1:25 $\pm$13,P:9$\pm$6 | F1:0$\pm$0,P:0$\pm$0 | F1:0 $\pm$0,P:0$\pm$0 |
| GNAF [ICLR'22] | F1:74$\pm$4,P:75 $\pm$ 8 | F1:33 $\pm$9,P:38$\pm$8 | F1:1.5 $\pm$ 2, P: 3 $\pm$ 2 | F1:0.1$\pm$0,P:0.1$\pm$0 |
| **AnomalySimpleton** | *F1:92$\pm$4, P:91$\pm$6* | F1:4$\pm$1,P:23$\pm$10 | F1:0$\pm$0, P:0$\pm$0 | F1:0$\pm$0,P: 0$\pm$0 |
| **SPIE-AD + SINDY*** | **Not applicable** | **Not applicable** | **F1: 78$\pm$12, P: 83$\pm$7** | **F1:77$\pm$9, P:81$\pm$6** |
| **SPIE-AD + LTCNN*** | **Not applicable** | **Not applicable** | **F1:84$\pm$11,P:85$\pm$9** | **F1: 82$\pm$4,P: 85$\pm$9** |

Figure 7: Snippet of SPIE-AD performance for zero-shot MTAD against recent MTAD works on benchmark datasets.

---

**Algorithm 2 U2Detect**($\{X_i\}_{i=1}^W$,$\rho(.,.)$,$L$,$\sigma$,$d$,$\Omega$)

1: **input** Test data $\{X_i\}_{i=1}^W$ with U2, robustness function $\rho$, SMR function $L$, mean robustness $\sigma$, interval $d$ from CRIE algorithm, and $\Omega$ set of all coefficients recoverd from training set.
2: **output** U2 label
3: $\omega_i = L((X_i) : i \in 1 \dots W)$
4: Compute residual $R_i = \rho(\omega_i, \Omega) - \sigma$
5: **if** $R_i \in [\sigma - d, \sigma + d]$ **then**
6:    mark all samples in the window $X_i$ as 0 (not U2)
7: **else**
8:    mark all samples in the window $X_i$ as 1 (U2)
9: **end if**
10: **return** U2 label

---

## C PERFORMANCE INFLATION

Point adjustment strategy inflates performance.

## D LESSONS LEARNT WITH ANOMALYSIMPLETON

Table 6 (in Supplement) and 7 (in Supplement) shows AnomalySimpleton could utilize PA and data leakage to beat GANF Zhao et al. (2022) and USAD Audibert et al. (2020a) baselines on all real benchmark datasets and was on par with Anomaly Transformers Xu et al. (2022). However, when PA was eliminated, its F1 score drastically dropped. Moreover, if data leakage was disabled, then its F1 score became 0. This shows a worse case machine with very poor realistic performance can result in a very good anomaly detection method through the usage of point adjustment and threshold learning using test data. Through this misadventure, we have learned the following lessons:

**Lesson 1:** anomaly detection works should show results for both with / without PA or use metrics such as $PA\%K$ as proposed in Kim et al. (2022).

**Lesson 2:** anomaly detection works should explicitly address data leakage issue by either obtaining validation data from train set or ensuring that validation set and test set are mutually exclusive.

## E U2 SYNTHETIC DATASETS

*F8 Cruiser:* This is an aircraft pitch control system using a model predictive control for trajectory tracking. The U2 scenario is a **hardware failure in Introduction section** where the elevator gets jammed and maintains a constant position overriding the controller ($F8Stuck$). Another U2 scenario is the elevator responds slower than normal ($F8Slow$).

*UAV Altitude control:* This is a quadcoptor, whose altitude is controlled by four proportional integrative and derivative (PID) controllers. These controllers provide balanced thrusts in each propeller so that the UAV maintains a given height. The first U2 is a **software failure** that changes the gravity

parameter $g$ in the controller software ($UAVSimG$). The second U2 scenario is an **electromagnetic attack** on the UAV gyroscope sensor ($UAVEMA$).

*Automated insulin delivery system:* This is an hybrid close loop autonomous system that autonomously decides on insulin delivery for the most part, but requires **human intervention** with extra insulin delivery to manage meal intake. The human may trick the system to deliver a high dosage of insulin by announcing to the system that a large meal has been ingested without actually consuming the meal. This U2 scenario is called phantom meal ($AIDPhantom$). In the second scenario, the human participant poorly installs the insulin cartridge resulting in insulin occlusion or blockage. The block causes insulin build up and finally it gives way and injects an overdose of insulin $AIDCartridge$.

In all the U2 examples, U2 scenarios are generated at random times with random duration of U2 activation sampled from a distribution.

### E.1 WHY THESE DATASET ARE U2 AND NOT ANOMALIES?

**F8 Cruiser (hardware-induced U2):** We simulate the F8 aircraft using the MATLAB SINDY-MPC model from:

https://github.com/eurika-kaiser/SINDY-MPC/tree/master/EX_FLIGHT_CONTROL_F8

*F8Stuck* The SINDY-MPC F8 cruiser uses a pseudo-random bit stream (PRBS) elevator input for pitch control. After 4000 samples of nominal behavior, we jam the actual elevator actuation within the dynamical solver at its last valid value, while still logging the original PRBS stream. Thus, the recorded actuator signal looks normal, but the true actuator entering the dynamics is incorrect—an unmeasured hardware failure. F8Slow Similarly, after 4000 samples, the actuator entering the dynamics is reduced to 75% of the nominal PRBS value, but the logged PRBS input remains unchanged.

*Why these are U2:* In both cases the marginal distribution of recorded data is unchanged (Figure 2B), but the underlying actuation delivered to the system is different, altering the governing dynamics and producing a non-stationary process. This satisfies the definition of U2: same marginals, different underlying process.

**Quadcopter UAV (software and sensor-induced U2)** We use the quadcopter simulator from:

https://github.com/bobzwik/Quadcopter_SimCon

*UAVSimG — stealthy software attack* At 5 s we increase gravitational acceleration gfor 5 s. The controller, unaware of this manipulation, compensates with slightly larger rotor speeds (still within normal rotor limits), so the recorded sensor traces retain their usual marginal distribution. However, the physical model evolves under a different gravity constant.

*UAVEMA — barometer interference:* We simulate the presence of a strong magnet near the UAV. This subtly corrupts barometer calibration and yields small but persistent elevation errors. Such calibration faults mirror real helicopter crashes where barometer drift caused catastrophic outcomes.

**Why these are U2:** The recorded data remains marginally indistinguishable from normal (Figure 2B), but the system evolves under incorrect physics (wrong g) or incorrect altitude sensing, producing a non-stationary process with unchanged marginals.

**Automated Insulin Delivery (AID) System — human error + hardware fault U2:** We use the FDA-accepted Type 1 Diabetes simulator for glucose–insulin dynamics:

https://pmc.ncbi.nlm.nih.gov/articles/PMC4454102/

*AIDPhantom — phantom meal declaration:* A meal of 15 g carbohydrate is reported to the controller but not actually ingested. The sensed glucose values remain high but stable (as is typical when patients announce meals at high glucose values), so the marginal glucose distribution does not shift. However, the controller interprets the phantom meal as a rising-glucose scenario and becomes unnecessarily aggressive in insulin delivery despite the absence of any physiological post-meal rise. This creates a mismatch: the recorded data looks normal, but the controller's behavior—and therefore the closed-loop glucose–insulin dynamics—no longer matches the intended governing model.

*AIDCartridgeS — insulin pooling and burst delivery:* Here the recorded insulin-delivery request is correct (e.g., B units over t minutes), but the cartridge mechanically fails to deliver at the correct rate. Instead it delivers only 0.1B initially, pools insulin internally, and then releases the remaining 0.8Bt in a single burst once a threshold is reached. The logged insulin traces still appear normal because they reflect commanded delivery, not the actuator fault, so there is no marginal distribution drift. But the true physiological dynamics experienced by the patient model change dramatically because insulin is delivered in a physiologically incorrect pattern. This behavior is consistent with real-world failures reported in Medtronic case studies.

*Why these are U2 (and not anomaly):* In both AIDPhantom and AIDCartridgeS, the recorded data shows no distributional shift—glucose values remain within expected ranges and logged insulin traces look nominal. However, the underlying closed-loop governing equation changes:

1) In AIDPhantom, the controller becomes aggressive out of context, producing a different insulin–glucose dynamic model even though the sensed glucose shows no rising-meal pattern.

2) In AIDCartridgeS, the true insulin delivery pattern diverges substantially from the logged (nominal) delivery, changing the underlying physiological dynamics while the recorded marginals remain stable.

Thus both scenarios satisfy the definition of U2: no marginal distribution shift but a structural change in the underlying process, induced by human error (phantom meal) or actuator failure (insulin pooling).

## E.2 WHY REAL WORLD DATA ARE U2?

(a) Real-world U2 datasets: Medtronic R and Epilepsy R

These datasets represent true U2 events, i.e., situations where the underlying governing process changes but the marginal distribution of the sensed data does not.

**Medtronic R (AIDCartridge U2):** This dataset contains insulin-cartridge delivery failures where the observed (logged) insulin traces remain marginally normal, but the true physiological insulin delivery pattern changes dramatically due to pooling and burst-release. This alters the closed-loop insulin–glucose dynamics (non-stationary underlying process) without producing marginal distribution drift. This is exactly a U2 event.

**Epilepsy R:** Sudden seizure onset produces an acute transition in brain neuroplasticity and changes the intrinsic neural dynamics. This is a structural shift in the governing equation of the EEG process, i.e., a non-stationary change in the underlying generative model. However, seizure onset happens rapidly and may not manifest as a marginal distribution shift within a short detection window. This again matches the definition of U2: process-level change, no marginal drift. Thus, both Medtronic R and Epilepsy R are genuine U2 datasets, not anomalies.

(b) Real-world anomaly datasets

Because U2 events are rare, high-stakes, and often proprietary, large open datasets exhibiting true U2 behavior are extremely scarce. Therefore, we also evaluate on standard anomaly-detection datasets, which do exhibit marginal distribution shifts. Our method remains applicable here for the following reason:

i) In anomaly scenarios, marginal drift causes the estimated governing equation to differ from the normal model.

ii) Since our framework evaluates deviations in the recovered underlying model, it naturally detects anomalies as well—although this is not the primary focus of the paper.

This provides a broader empirical validation and enables comparison against widely used anomaly-detection baselines.

## E.3 LTC-NN MODEL RECOVERY ROBUSTNESS RESULTS

Table S1 5 shows the performance of LTC-NN architecture described in Fig. 8 of the main pa-

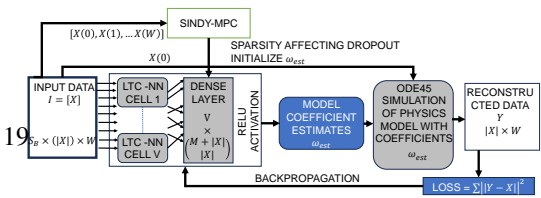

per on model recovery for different benchmark examples available in Kaiser et al. (2018).

For each evaluation experiment, we use two metrics:

**Root mean square error in model coefficients** ($RMSE_\Theta$) and **Root mean square error in signal** ($RMSE_Y$). Given the estimated model coefficients $\Theta_{est}$ and measured variables $Y_{est}$ for any technique we computed them as:

$$RMSE_\Theta = \sqrt{\frac{1}{p} \sum_{j=1\ldots p} (\Theta_{est}^j - \Theta^j)^2}, \tag{3}$$

$$RMSE_Y = \frac{1}{n} \sum_{l=1\ldots n} \sqrt{\frac{1}{k} \times \sum_{j=1\ldots k} (Y_{est}^l(j) - Y^l(j))^2}. \tag{4}$$

Table 5: S1: Comparison of LTC-NN architecture with baseline SINDY-MPC only and other RNN architectures on standard benchmarks. LTC-NN-MR represents model recovery with LTC-NN architecture shown in Fig. 8. The LTC-NN can be replaced by CT-RNN or NODE. Value in () is standard deviation

| Example | RMSE | SINDY-MPC | LTC-NN-MR | CT-RNN-MR | NODE-MR |
|---|---|---|---|---|---|
| Lotka | $RMSE_\Theta$ | 0.059 (0.02) | 0.048 (0.015) | 0.054 (0.03) | 0.064 (0.02) |
| Volterra | $RMSE_Y$ | 0.03 (0.02) | 0.03 (0.018) | 0.05 (0.02) | 0.088 (0.03) |
| Chaotic | $RMSE_\Theta$ | 0.014 (0.008) | 0.015 (0.006) | 0.022 (0.009) | 0.044 (0.012) |
| Lorenz | $RMSE_Y$ | 1.7 (0.6) | 1.68 (0.4) | 3.66 (1.1) | 8.1 (3.6) |
| F8 | $RMSE_\Theta$ | 7.9 (3.2) | 6.8 (2.9) | 10.5 (4.8) | 19.9 (7.4) |
| Crusader | $RMSE_Y$ | 3.2 (2.1) | 1.57 (1.4) | 3.46 (2.6) | 7.22 (5.7) |
| Pathogenics | $RMSE_\Theta$ | 0.5 (0.2) | 0.39 (0.23) | 0.43 (0.3) | 0.42 (0.3) |
| attack | $RMSE_Y$ | 27.8 (9.1) | 28.3 (6.2) | 28.8 (7.7) | 29.5 (9.6) |

### E.4 DESCRIPTION OF REAL WORLD DATASETS

We used three real datasets:

**Server Machine Database:** The Server Machine Dataset (SMD) is a newly curated dataset that spans a period of five weeks, collected from a major Internet company known for its extensive server infrastructure Su et al. (2019). This dataset, which includes detailed logs and metrics related to server machine performance, has been made publicly available on GitHub to support research in anomaly detection and related fields.

The SMD dataset comprises a wide range of features, including CPU utilization, memory usage, disk I/O, and network traffic, collected at regular intervals. For practical analysis, we have divided the dataset into two equal-sized subsets: the first subset, which covers the initial period of the data collection, is used as the training set. The second subset, covering the remaining period, is designated as the testing set.

In the testing subset, domain experts have meticulously identified and labeled anomalies, along with their specific dimensions, based on a thorough examination of incident reports and historical data. These labels provide valuable insights for evaluating anomaly detection algorithms and enhancing their accuracy.

**Soil Moisture Active Passive Satellite:** The Soil Moisture Active Passive (SMAP) satellite Liu et al. (2024) is a NASA mission designed to measure and monitor soil moisture levels across the globe. SMAP employs a combination of active radar and passive radiometer technologies to provide high-resolution measurements of soil moisture, which are crucial for understanding water cycles, weather patterns, and climate change. The satellite records key performance indicators (KPIs) related to its operational status and performance metrics, including data on the satellite's health, instrument functionality, and environmental conditions. These KPIs are essential for ensuring the proper functioning of the spacecraft and for diagnosing and addressing any issues that may arise during its mission.

**Mars Science Laboratory Rover (MSL):** The Mars Science Laboratory (MSL) rover Liu et al. (2024), commonly known as Curiosity, is a NASA rover mission designed to explore the surface of

Table 6: Comparison of MTAD methods on real-world datasets (F1; Time in minutes). Left block: SMD, middle: SMAP, right: MSL.

| MTAD Method | SMD | | | | SMAP | | | | MSL | | | |
|---|---|---|---|---|---|---|---|---|---|---|---|---|
| | A3 | ¬A3 | ¬A2 | *Time* | A3 | ¬A3 | ¬A2 | *Time* | A3 | ¬A3 | ¬A2 | *Time* |
| AT | 90.7 | 38.8 | 0 | 372 | 91.2 | 22.3 | 0 | 183 | 88.6 | 13.1 | 0 | 175 |
| GANF | 78.6 | 41.2 | 3.4 | 361 | 71.9 | 32.8 | 1.1 | 179 | 73 | 24 | 0 | 165 |
| USAD | 43.1 | 21.2 | 21.2 | 218 | 62 | 26 | 26 | 121 | 41 | 18 | 18 | 103 |
| OFA | 72.9 | 2.5 | 1.9 | 318 | 86.9 | 9.4 | 5.1 | 171 | 82.7 | 22.3 | 4.4 | 159 |
| FITS | 99.9 | 32.7 | 11.2 | 281 | 70.74 | 13.4 | 2.2 | 164 | 78.12 | 15.3 | 4.3 | 141 |
| TFAD | 89.3 | 21.7 | 4.1 | 211 | 96.3 | 35.4 | 7.7 | 135 | 96.4 | 40.1 | 8.8 | 122 |
| AnomalySimpleton | 96.2 | 2.0 | 0 | 21 | 90.5 | 4 | 0 | 7 | 89.5 | 4.8 | 0 | 6 |
| SPIE-ADS | 74 | 73 | 73 | 172 | 68 | 65 | 65 | 153 | 83 | 83 | 83 | 132 |
| SPIE-ADL | 86 | 86 | 86 | 323 | 79 | 73 | 73 | 208 | 83 | 83 | 83 | 178 |

Table 7: Comparison of AD methods on univariate datasets (event-wise accuracy).

| AD Method | UCR | Yahoo | NAB |
|---|---|---|---|
| MatrixProfile | 0.512 | 0.23 | 0.2 |
| AT | 0.4 | 0.2 | 0.78 |
| TimeVQVAE | 0.708 | 0.4 | 0.6 |
| TranAD | 0.19 | 0.6 | 0.92 |
| OFA | 0.5 | 0.8 | 0.92 |
| FITS | 0.47 | 0.8 | 0.9 |
| TFAD | 0.37 | 0.8 | 0.6 |
| AnomalySimpleton | 0.13 | 0.2 | 0.2 |
| SPIE-ADS | 0.756 | 0.8 | 0.92 |
| SPIE-ADL | 0.756 | 0.8 | 0.94 |

Mars. Equipped with a suite of scientific instruments, the MSL rover conducts a variety of experiments to study Mars' geology, climate, and potential for past habitability. The rover records KPIs related to its operational performance, such as power consumption, temperature readings, and communication status. These performance metrics are critical for monitoring the health and functionality of the rover, managing its systems, and troubleshooting any technical challenges that arise during its exploration of the Martian surface. The data collected helps scientists and engineers ensure the rover's effective operation and mission success.

### E.5 EXTENDED TABLE FOR REAL WORLD DATASET

Table S2 8 shows the extended results for Table 6 and 7 in supplement with precision and recall values.

### E.6 SPIE-AD HYPER-PARAMETER OPTIMIZATION

Given a threshold of $r\%$, the hyper parameters of the SPIE-AD method extracts the hyper-paramters of the SPIE-AD method so that atleast $r\%$ data from the training set falls within the robustness interval $[\rho_1, \rho_2]$, while minimizing $(\rho_2 - \rho_1)$. The algorithm currently is a brute force search through all possible hyper-parameter combination to find the best hyper-paramters that matched the above-mentioned conditions.

### E.7 MORE INFORMATION ON DISTRIBUTION SHIFT

Although the definition of anomaly does not directly imply a distribution shift between anomalous and normal data, analysis of existing anomaly benchmark datasets reveal otherwise (Table 9). We use the Kolmogorov-Smirnov (KS) hypothesis test (KS, 2008) to evaluate difference in distribution parameters between normal and anomalous data. It is observed that in almost all real world benchmark datasets $> 90\%$ of test cases have data distribution shift. In Table 9 we report percentage of distribution shift (PDS) the percentage of anomalous data which has a different distribution than normal data as given by the KS test.

### E.8 THRESHOLD INDEPENDENT METRICS

SPIE-AD is not a anomaly thresholding model. The only control knob we have is the converage level $\alpha$ in Algorithm CRIE. In all our experiments it is set to 0.05 which is standard in most statistical methods. As such $\alpha$ is not a threshold but it has a role in determining the robustness range. To obtain AUC ROC and AUC PR, VUS ROC and VUS PR, we varied this $\alpha$ from 0.05 to 0.2 in steps of 0.025. In this regard I used the code available in https://github.com/TheDatumOrg/VUS to compute all the abovementioned metrics of some of the baselines in Table 3 and 4. In particular we took the top 3 baselines and SPIE-ADS from Table 3 to get the Table 10.

Table 8: S2: Comparison of SPIE-AD with latest baseline techniques on real world datasets and ablation studies. The datasets all satisfy A1.

| Method | SMD | | | | | | | | |
|---|---|---|---|---|---|---|---|---|---|
| | A3 | | | ¬ A3 | | | ¬ A2 | | |
| | Pr | Re | F1 | Pr | Re | F1 | Pr | Re | F1 |
| AT | 83 | 100 | 90.7 | 29 | 58.6 | 38.8 | 0 | 0 | 0 |
| GANF | 39.5 | 93 | 78.6 | 28 | 78 | 41.2 | 30.6 | 1.8 | 3.4 |
| USAD | 28 | 94 | 43.1 | 12.2 | 80 | 21.2 | 12.2 | 80 | 21.2 |
| AnomalySimpleton | 98.2 | 94.4 | 96.2 | 35.1 | 1.0 | 2.0 | 0 | 0 | 0 |
| SPIE-ADS | 64 | 87.7 | 74 | 63 | 86.7 | 73 | 63 | 86.7 | 73 |
| SPIE-ADL | 84 | 88 | 86 | 83 | 89 | 86 | 83 | 89 | 86 |
| Method | SMAP | | | | | | | | |
| | A3 | | | ¬ A3 | | | ¬ A2 | | |
| AT | 83.8 | 100 | 91.2 | 12.7 | 90 | 22.3 | 0 | 0 | 0 |
| GANF | 57.5 | 96 | 71.9 | 19.9 | 93 | 32.8 | 0.6 | 7 | 1.1 |
| USAD | 45 | 100 | 62 | 15.1 | 94 | 26 | 15.1 | 94 | 26 |
| AnomalySimpleton | 86.4 | 95.1 | 90.5 | 13.6 | 2.4 | 4 | 0 | 0 | 0 |
| SPIE-ADS | 55 | 89 | 68 | 52 | 87 | 65 | 52 | 87 | 65 |
| SPIE-ADL | 69.8 | 91 | 79 | 65.7 | 82.1 | 73 | 65.7 | 82.1 | 73 |
| Method | MSL | | | | | | | | |
| | A3 | | | ¬ A3 | | | ¬ A2 | | |
| AT | 79.5 | 100 | 88.6 | 8.7 | 27 | 13.1 | 0 | 0 | 0 |
| GANF | 64 | 85 | 73 | 16 | 48 | 24 | 0 | 0 | 0 |
| USAD | 44.5 | 38 | 41 | 14.5 | 23.8 | 18 | 14.5 | 23.8 | 18 |
| AnomalySimpleton | 89.6 | 89.4 | 89.5 | 20.9 | 2.7 | 4.8 | 0 | 0 | 0 |
| SPIE-ADS | 80.2 | 86 | 83 | 80.2 | 86 | 83 | 80.2 | 86 | 83 |
| SPIE-ADL | 80.3 | 85.8 | 83 | 80.3 | 85.8 | 83 | 80.3 | 85.8 | 83 |

| Dataset | No. of time series | PDS |
|---|---|---|
| Server Machine Dataset Su et al. (2019) | 38 | 91% |
| Soil Moisture Active Passive Su et al. (2019) | 25 | 92% |
| Mars Science Lab Rover Su et al. (2019) | 55 | 93% |
| SWaT dataset Goh et al. (2017) | 51 | 92% |
| WADI dataset Ahmed et al. (2017) | 123 | 95% |
| Pooled Server Metric Abdulaal & Lancewicki (2021) | 25 | 100% |
| UCR anomaly detection Wu & Keogh (2023) | 250 | 94% |
| Yahoo anomaly detection Yoshihara & Takahashi (2022) | 100 | 91% |
| NAB dataset Ahmad et al. (2017) | 58 | 94% |

Table 9: S3: Percentage of distribution shift (PDS) in anomaly detection benchmark datasets

Table 10: S4: Threshold independent metrics

| Method | SMD AUC ROC | SMD AUC PR | SMD VUS ROC | SMD VUS PR | SMAP AUC ROC | SMAP AUC PR | SMAP VUS ROC | SMAP VUS PR | MSL AUC ROC | MSL AUC PR | MSL VUS ROC | MSL VUS PR |
|---|---|---|---|---|---|---|---|---|---|---|---|---|
| AT | 0.48 | 0.46 | 0.59 | 0.56 | 0.37 | 0.35 | 0.44 | 0.41 | 0.29 | 0.25 | 0.33 | 0.29 |
| FITS | 0.57 | 0.55 | 0.62 | 0.59 | 0.41 | 0.37 | 0.47 | 0.43 | 0.42 | 0.38 | 0.47 | 0.44 |
| TFAD | 0.60 | 0.55 | 0.64 | 0.58 | 0.43 | 0.41 | 0.47 | 0.44 | 0.46 | 0.44 | 0.53 | 0.50 |
| SPIE-ADS | 0.61 | 0.58 | 0.65 | 0.61 | 0.52 | 0.51 | 0.57 | 0.54 | 0.62 | 0.58 | 0.64 | 0.59 |

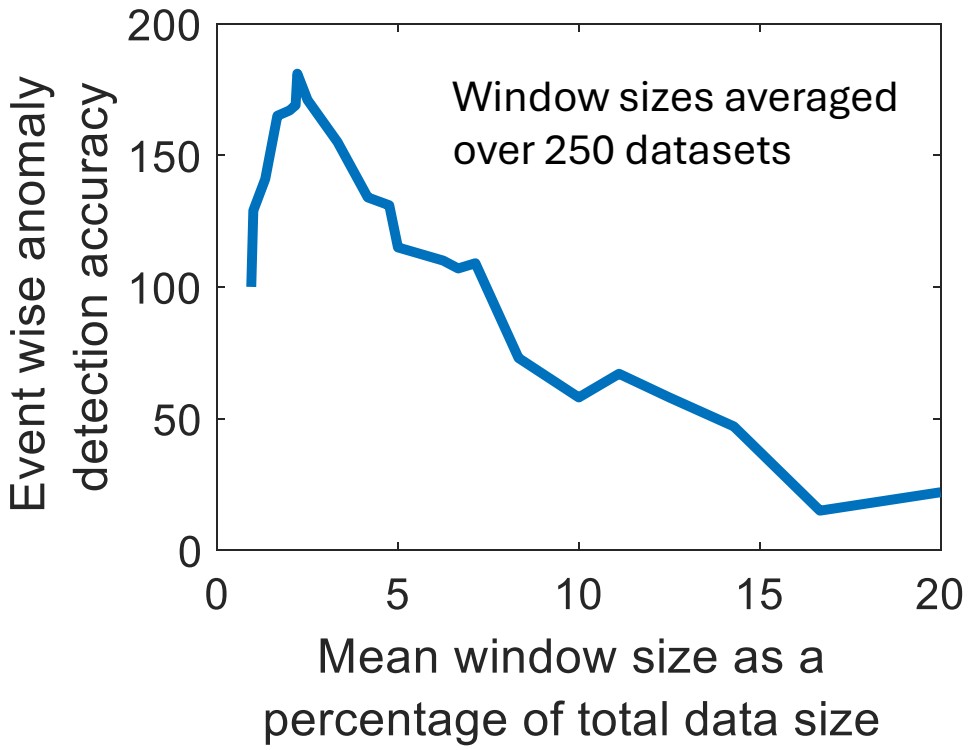

Figure 9: Event wise anomaly detection accuracy of SPIE-ADS with varying window size. Results averaged over 250 UCR datasets.

## F WINDOW SIZE SENSITIVITY

We use the UCR database to evaluate sensitivity to window size for our approach as it has the largest number of real world datasets (n = 250) to ensure statistically stable results. The window size is varied as a percentage of the total dataset size for each database. Fig. 9 shows that large window sizes reduces the accuracy of detecting an anomalous event since the event size maybe a small fraction of the window size. When the window size is too small, SINDY-MPC core fails to extract accurate models of the underlying governing dynamics - decreasing its accuracy. Hence, there is a optimal window size for each dataset.

## G COMPUTATIONAL COMPLEXITY

There are two MR cores of SPIE-AD: SINDY-MPC and LTC-NN. SINDY-MPC uses the sequential threshold ridge regression (STRidge) Kaiser et al. (2018) strategy. The computational complexity of Ridge regression in the worst case is O($Nn^2$), for $N$ samples and $n$ dimensions, since for MTAD number of regularization parameters is less than $N$ Wang & Pilanci (2023). The sequential threshold runs Ridge regression multiple times until a desired reconstruction accuracy is obtained. If we fix a maximum $Q$ number of iterations then the overall computational complexity of SINDY-MPC is O($QNn^2$). For the LTC-NN architecture, the computation complexity of forward pass is $O(V + V(|\Theta| + q)) + O(|X|N)$, where, $V$, $q$, $\Theta$, $X$ are as in Fig. 8. Complexity of backward pass is $O(VP_{LTC}N + V(|\Theta| + q)P_{dense}N)$, where $P_{LTC}$ is the number of parameters in the LTC cell, and $P_{dense}$ is the number of parameters in each neuron of the dense layer. SINDY-MPC on a single CPU thread was 11.3 ($\pm$ 2.1) times faster than LTC-NN on GPU. The overall computational complexity is $O((N/W)QNn^2)$ for SPIE-ADS and $O((N/W)VP_{LTC}N + V(|\Theta| + q)P_{dense}N)$.

Table 11: Related works. Sampling High is > Nyquist rate, Low is = Nyquist rate. Bold = baselines.

| Approach | Sampling | Inputs | Rationale for baseline |
|---|---|---|---|
| SINDy Quade et al. (2018) | High | No | Cannot handle inputs |
| **SINDy-MPC** Kaiser et al. (2018) | **High** | **Yes** | **Widely used** |
| E-SINDy Fasel et al. (2022) | Low | No | SINDy-MPC is E-SINDy + inputs |
| **W-SINDy** Messenger & Bortz (2021) | **High** | **No** | **Focuses on noise reduction** |
| **LTC-NN (This Work)** | **Low** | **Yes** | **Proposed in this paper** |

Accuracy, Detection delay, Precision, Event Coverage, False Acceptance Rate, and F1 score for high noise 5dB signal to noise ratio for SPIE-ADL

Figure 10: Performance variance with respect to sampling frequency, and library size (higher polynomial order results in combinatorially larger library). High Noise case. Low noise case in main paper Fig. 3.

# H   ADDITIONAL EXPERIMENTS

Table 11 shows different SINDY variants.

## H.1   DATA AND CODE AVAILABILITY

The data and code for model recovery using SINDY-MPC are available in `https://github.com/ImpactLabASU/U2Recognition`

To use LTC-NN a manual transfer of model coefficient is required and the pipeline is not entirely automated. Hence, the models available in `https://github.com/ImpactLabASU/LTC-NN-MR` has to be run first and the saved model coefficients needs to be transferred to the U2Recognition github and then run the files described in the U2Recognition github.

The AnomalySimpleton also known as SMDTrash is available in `https://github.com/ImpactLabASU/AnomalyAbsurd`

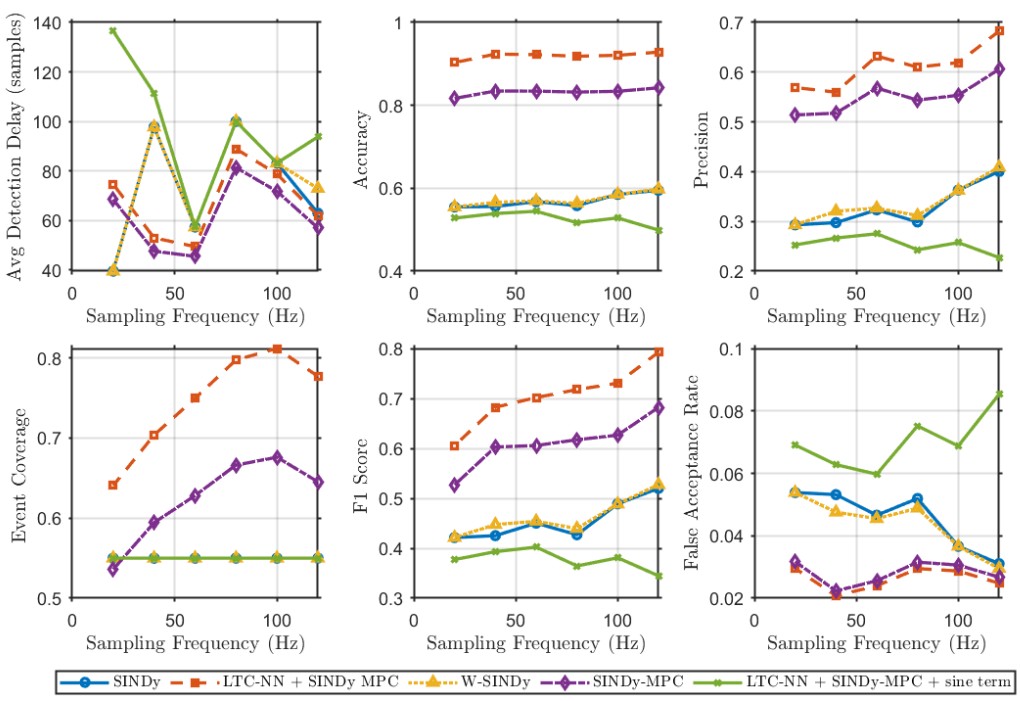

Figure 11: Effect of different SINDY variants on SPIE-AD.

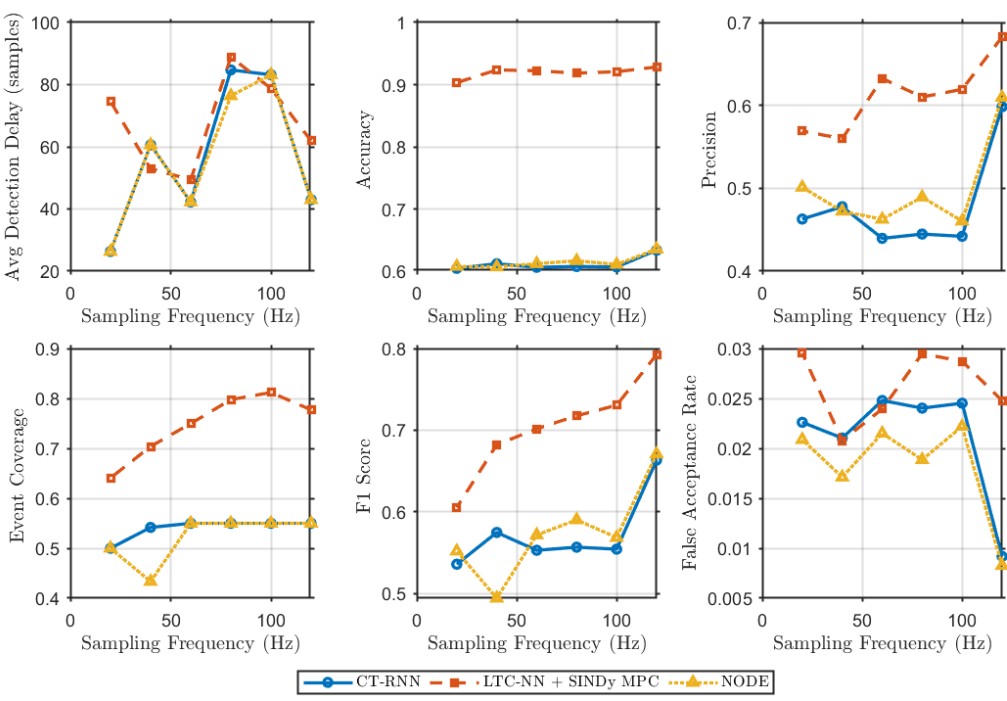

Figure 12: Effect of different continuous time backbones on SPIE-AD.

