# OpenReview forum: "Detection of unknown unknowns in autonomous systems"
_ICLR.cc/2026/Conference — ICLR 2026 Poster_

### Official Review · Reviewer_16mj · 2025-10-18

**Soundness:** 4
**Presentation:** 3
**Contribution:** 3
**Rating:** 8
**Confidence:** 4

**Summary:**

This paper tackles the emerging problem of unknown-unknowns (U2s) in time-series anomaly detection—cases where there is no obvious out-of-distribution shift from normal data, making traditional detection methods ineffective. The authors identify three key challenges: the absence of distributional deviation, the impractical enhancement of existing models, and data leakage in threshold calibration. To address these, they propose a sparse model identification–enhanced anomaly detection framework tailored to handle U2 scenarios. Extensive experiments are conducted on synthetic U2 benchmarks and multiple real-world multivariate anomaly detection datasets, showing that the proposed method outperforms a range of baselines.

**Strengths:**

The paper explores an interesting and relatively new direction in anomaly detection by formally framing and tackling the concept of unknown-unknowns. The writing is generally clear, and the empirical section is comprehensive and convincing. The authors provide strong evidence that their method can distinguish subtle anomalies even when there is no visible distribution shift. The experiments are rigorous, covering both synthetic U2 datasets and widely used multivariate benchmarks, with appropriate metrics and detailed comparisons. Overall, the work is solid, timely, and contributes a valuable perspective to the time-series anomaly detection literature.

**Weaknesses:**

1. While the technical contribution and empirical results are strong, the paper could be significantly improved in presentation and positioning. The current layout is too compact: equations appear small and dense, while several tables (e.g., Tables 1, 3, and 4) occupy disproportionate space, making the interleaved text uneven and hard to read. Improving figure/table sizing, rounding consistency, and overall margin balance would greatly enhance readability. Adding visual highlights or color emphasis in key tables could also help.

2. The related-work discussion is incomplete and occasionally overstated. The statement “To the best of our knowledge, there is only one solution for zero-shot MTAD Audibert et al. (2020)” is inaccurate. Other relevant works, such as ITF-TAD, UniTS, should be properly discussed and contrasted. Also, the paper only mentions SIGLLM but omits relevant methods such as VLM4TS, which also address zero-shot and training-free detection in related settings. A fairer and more complete comparison would strengthen the contribution.

3. Beyond metrics, it would also be valuable to include additional qualitative analyses or visual case studies. For instance, Figure 2 could be expanded to show how different methods (including baselines) behave under normal versus anomalous regimes, visually demonstrating how the proposed model differentiates subtle shifts that are not captured by these metrics.

Namura, Nobuo, and Yuma Ichikawa. "Training-free time-series anomaly detection: Leveraging image foundation models." arXiv preprint arXiv:2408.14756 (2024).
Gao, Shanghua, et al. "Units: A unified multi-task time series model." Advances in Neural Information Processing Systems 37 (2024): 140589-140631.
Alnegheimish, Sarah, et al. "Large language models can be zero-shot anomaly detectors for time series?." arXiv preprint arXiv:2405.14755 (2024).
He, Zelin, Sarah Alnegheimish, and Matthew Reimherr. "Harnessing Vision-Language Models for Time Series Anomaly Detection." arXiv preprint arXiv:2506.06836 (2025).

**Questions:**

Could the authors provide quantitative or qualitative evidence illustrating how their method distinguishes U2 anomalies?

As the authors note that SPIE-AD struggles with short-lived anomalies, could the authors elaborate on potential approaches to alleviate it?

---

> ### Author Response · Authors · 2025-11-22
> **Thank you for your thorough comments**
>
> We are running more experiments suggested by you as well as Reviewer u3Fj. It is taking longer than expected. We will update the paper addressing all the reviews before November 24th.

---

> ### Author Response · Authors · 2025-11-27
> **Revised Version Uploaded**
>
> Thank you for your extensive reviews and suggestions regarding readability of the paper as well as discussion on related work. We have updated the paper according to your comments. In summary we have done the following to address your comments:
>
> **a)** Changed Figure 2 to make it Figure 1 in updated paper that shows an example of U2, F8 Cruiser Stuck elevator. Here we show the state variables changing over time before and after U2. It clearly shows that U2 does not result in any marginal distribution change. We then show the underlying model extracted using SPIE-AD, which shows significant non-stationary change. We then show an example baseline USAD KDD 2020 to show that anomaly score from exemplary MTAD approaches do not change significantly under U2 events. We show that if validation data is obtained from training data then the anomaly score threshold does not even detect the U2 event. If data leakage is permitted then it does detect the U2 event but the anomaly score threshold is suboptimal and results in significant false positives.
>
> **b)** Added a paragraph on why few / zero shot anomaly detection techniques do not apply for U2 detection. This paragraph discusses all papers that you listed in your comments.
>
> **c)** Moved some less critical tables to appendix and converted some of them to figures.
>
> **d)** We added several new experiments to address comments regarding sensitivity of SPIE-AD to changes in various hyperparameters in response to Reviewer u3Fj
>
> **e)** Updated several areas of the write up in response to Reviewer H3ew.
>
> Now we address each of your comments in detail.
>
> ## Q1: related work discussion is incomplete and overstated ##
>
> We have updated Introduction and related works to directly address the related work that you have suggested. Please refer to Lines 106 to 113.
>
> **UniTS Gao et al** - To the best of our comprehension, UniTS is a few shot learning mechanism which requires fine tuning LLM with small number of examples. This is indicated in Appendix C2 of UniTS paper where the authors write "The output for sample tokens zx is unpatchified to obtain the predicted sample ˆx. During inference, following the approach in [112], we determine a threshold of reconstruction error from the training and testing data, which is then used to detect anomalous time series points. " This implies that anomaly examples are used to tune reconstruction error thresholds.
>
> Further looking at their code available in https://github.com/mims-harvard/UniTS/blob/main/exp/exp_sup.py  we observe the following:
>
> a) test data is used for threshold tuning lines 786 to 806 in exp_sup.py
>
> b) point adjustment used for updating anomaly detection results lines 818 to 824 in exp_sup.py.
>
>
> In U2 scenario, we have no examples, we are attempting zero shot detection which UniTS does not do for anomaly task. Nevertheless, we can still compare UniTS with SPIE-AD on the anomaly detection benchmarks SMD MSL and SMAP. For SMD, MSL, SMAP, UniTS gets an F1 score of 88\%, 83\%, 83\% with threshold tuning using test data (few shot + potential data leakage), SPIE-AD gets 86\%, 83\%, 79\% without threshold tuning using test data. Moreover Units results are reported with point adjustment.  For these reasons we did not include UniTS results in our paper.
>
> **Namura et al ITF-TAD** This paper converts the time series into images and then uses image based foundational models. In typical deployments of U2 we assume resource constrained systems with strict memory restrictions. Hence such image conversion tools may not be available and may be resource (memory) hungry. Hence we did not include this paper as a comparator in our main paper. For the same reason Zelin et als 2025 paper on "Harnessing Vision-Language Models for Time Series Anomaly Detection" was not included as a comparator. Moreover, Zelin et al is tested only on univariate data which will not conform to U2 detection since it requires multivariate data. Similarly, Alnegheimish, Sarah, et al. "Large language models can be zero-shot anomaly detectors for time series?." is also tested on univariate data.  All our comparators used just the time series in signal domain and processed multi-variate data.
>
> Nevertheless, we can still compare with ITF-TFAD on UCR data where SPIE-AD outperforms getting 189 correct (Table 7 in Appendix. since this is univariate anomaly detection) v.s. 150 for ITF-TFAD. SPIE-AD also outperforms ITF-TFAD on SMD, SMAP and MSL datasets.
>
>
> ## Q2: quantitative or qualitative evidence illustrating how their method distinguishes U2 anomalies ##
>
> Following your suggestion, we have updated the Figure 2 which is now Figure 1 and it discusses a clear example of how our method distinguishes U2 and how an exemplary zero shot MTAD method USAD fails to detect U2 for the elevator stuck U2 in F8 cruiser. The main distinction is that our method extracts the underlying model and can hence detect non-stationary changes which existing MTAD methods struggle to detect.

---

### Official Review · Reviewer_H3Ew · 2025-10-31

**Soundness:** 2
**Presentation:** 2
**Contribution:** 2
**Rating:** 2
**Confidence:** 3

**Summary:**

This work discusses the problem of operating in the presence of unknown unknowns (U2) data from multivariate time series, based on the hypothesis that state-of-the-art anomaly detection mechanisms are not well-suited to identify this type of data, when any of three assumptions are violated:

1- There is a shift in the data distribution due to an anomaly \
2- Data leakage is used to fine-tune the anomaly threshold hyperparameter \
3- The evaluation setup takes place in an unrealistic setup (i.e, anomalies are continuous)

The paper presents the following contributions:
1) It introduces SPIE-AD, a methodology to detect U2 when the above mentioned assumptions do not hold. At its core, SPIE-AD consists of a sparse non-linear dynamical model recovery strategy, and model robustness interval extraction through conformal prediction. \
2) Six synthetic benchmarks s derived from U2 scenarios occurring in three different real-world system I am running a few minutes late; my previous meeting is running over.\
3) Experimental demonstration of the disadvantages of using point adjustment (PA)

**Strengths:**

- The problem of dealing with unknown unknowns is highly relevant in many applications, still challenging, and actively researched.
- The paper gathers a good number of baselines and datasets for their experiments

**Weaknesses:**

- The explanation/definition of U2 is based on a couple of examples illustrated in the paper. A formal (mathematical?) definition is highly desirable to clearly establish when a sample should be denoted as a U2 or an anomaly.
- Conformal prediction as per [19] is about estimating prediction intervals even when the distribution of the training and test sets deviate, under the condition that the training data is i.i.d.. This, however, does not hold strictly for time series as subsequent points are not independent. The paper does not specify under which circumstances the assumption can be relaxed in this work
- Some details about the multiple hyperparameters, terms involved, and precise function definitions (i.e., $\rho$) are not specified.

**Questions:**

At the core of the assumptions and hypotheses that drive this work is the definition of what constitutes an unknown unknown. This is not entirely clear from the paper, and in particular from the datasets used. Hence:

1. For the synthetic datasets: how exactly are the abnormal scenarios generated? Following the definitions in the paper, what makes them a U2 and not an anomaly?
2. For the real-world datasets: From the description, it seems that this work deals simply with anomalies. However, the introduction discusses that an anomaly is not a U2. What is your position here?

Overall, from the paper, it is not clear why the proposed method should be consider as an U2 detector, rather than a "classical" MTAD. Please develop.

---

> ### Author Response · Authors · 2025-11-13
> **Thank you for thoroughly reading our paper and your insightful comments**
>
> We address your comments, hope for a fruitful discussion.
>
> ## Q1: U2/anomaly definition ##
>
> Answer: An *anomaly* causes a window-level marginal distribution shift even when the underlying process is stationary, typically due to transient external perturbations that do not change the true generative mechanism.
>
> A *U2 event* preserves marginal distributions across windows but makes the underlying process non-stationary by altering the temporal dependence structure, typically due to unmeasured input violations, stealthy software attacks, or hardware failures.
>
> *Mathematically* let $X(t) $ be a stochastic process with global marginal distribution $P(x)$ and underlying dynamics $dX/dt = f(X,u,t,\theta(t))$, $\theta(t)$ is dynamics parameter at time t.
>
> A window $W$ is an **anomaly** if: a) **Window marginal drift occurs**: $P_t(x) \;\not\approx P(x),$ for some $t\in W$, as determined by a statistical test (Figure 2); b) **Underlying process remains stationary**: $\theta(t) = \theta(t+\tau), \forall t,\tau$; c) **Deviation arises from a transient external perturbation**: $X_t = X_t^{sys} + \eta_t,\eta_t \neq 0$ only for $t\in W$. Thus, an anomaly induces local marginal deviation without altering the long-term distribution.
>
> A window $W$ is a **U2 event** if: a) **Marginal distributions remain unchanged**: $P_t(x) \approx P(x), \forall t\in W$; but b) **Process becomes non-stationary**: $\theta(t) \neq \theta(t+\tau)$, for some $\tau$. Thus, a U2 event preserves marginal statistics but reflects a structural change in the system's underlying dynamics.
>
> ## Q2: Conformal prediction iid assumption ##
>
> Answer: We do not apply conformal prediction to the raw series (non-iid) but to the window-wise estimated governing equations. With no U2 event, dynamics are stationary, so each window estimates the same model, giving i.i.d. estimates up to noise. Thus iid applies to model estimates, not raw data, making conformal prediction valid.
>
> A U2 changes the governing equation, making post-event estimates non-iid and causing conformal scores to break bounds, triggering U2 detection.
>
> ## Q3: Synthetic data U2/Anomaly ? ##
>
> Answer: **F8 Cruiser** (SINDY-MPC model):
> https://github.com/eurika-kaiser/SINDY-MPC/
>
> **F8Stuck** F8 cruiser uses a PRBS elevator input for pitch control. After 4000 samples, we jam the elevator actuation in the dynamics at its last valid value while still logging the original PRBS stream. Thus the recorded input appears normal, but the true actuation is incorrect, an unmeasured hardware failure.
>
> **F8Slow** After 4000 samples, the actuator entering the dynamics is reduced to 75% of the nominal PRBS value, but the logged PRBS input remains unchanged.
>
> **Why U2?** No marginal distribution change (Fig. 2B), true actuation change alters governing dynamics and creating a non-stationary process, hence U2.
>
> **Quadcopter UAV** https://github.com/bobzwik/Quadcopter_SimCon
>
> **UAVSimG**: At 5 s we increase unmeasured controller setting, gravitational acceleration g for 5 s simulating software attack. The controller compensates with higher rotor speeds (within limits), so recorded sensor traces keep their usual marginals, while the physical model evolves under a different g.
>
> **UAVEMA**: We simulate the presence of a strong magnet near the UAV. This corrupts barometer calibration (unmeasured) and yields persistent elevation errors.
>
> **Why U2?** No marginal distribution change (Fig. 2B), but system evolves under incorrect physics (wrong g) or faulty altitude sensing, producing non-stationarity.
>
> **AID**: UVA/PADOVA T1DSim
>
> **AIDPhantom**: A phantom meal is reported but not ingested. The controller treats the phantom meal as a rising-glucose event and becomes aggressive with insulin, despite no post-meal glucose rise. Thus, the recorded data looks normal, but the controller’s actions, and the resulting dynamic, no longer match the true governing model.
>
> **AIDCartridgeS** The recorded insulin request is correct (B U over t mins), but the cartridge fails mechanically. It delivers only 0.1B initially, pools insulin, then releases 0.8Bt in a single burst once a threshold is reached. Logged traces look normal since they reflect commanded, not actual,  delivery, so marginals do not shift. However, the patient experiences a dramatically different insulin pattern.
>
> **Why U2?** Recorded data shows no marginal shift, yet the closed-loop dynamics change, hence U2.
>
> ## Q4: Real data U2/anomaly? ##
> (a) U2 datasets:
>
> Medtronic R: insulin-cartridge U2 scenario.
>
> Epilepsy R: seizure onset induces rapid neurodynamic shifts (change in EEG generative model) without clear marginal changes in short windows. Both are true U2 events.
>
> (b) Anomaly: True U2 datasets are rare and often proprietary, so we test on anomaly datasets. In anomalies, marginal probability changes cause the recovered governing equation to differ from normal for short windows making them detectable by our framework even though anomaly detection is not our primary goal.

---

> > ### Author Response · Authors · 2025-11-13
> > **Regarding hyper parameters**
> >
> > We acknowledge your concerns. We have attempted to include several hyperparameters in our supplementary documents including data generation code and SPIE-AD execution code. If you can kindly let us know which hyper-parameters you are interested in we can either direct you to the appropriate place in supplementary document or provide you with new information.

---

> > > ### Author Response · Authors · 2025-11-27
> > > **Revised version uploaded**
> > >
> > > Dear Reviewer,
> > >
> > > Thank you for your comments again. We have significantly updated our paper to address all reviewer comments. In summary we have done the following changes:
> > >
> > > a) Updated Figure 1 to better explain the difference between U2 and Anomalies and why our technique achieves superior U2 detection performance. The main reason is U2 induces non-stationary behavior which SPIE-AD can detect since it is extracting  underlying dependency structures of the multi-variate data.
> > >
> > > b) Added a mathematical definition of U2 in Section 2.
> > >
> > > c) Performed extensive new experiments for different hyper-parameter changes and ablations.
> > >
> > > You comments are addressed in the following sections of the paper:
> > >
> > > **Q1:** We provide a mathematical definition of U2 clearly distinguishing from anomaly. This is depicted in Figure 1, and in the write up in Lines 46 - 50, 59 - 64, 183 - 199. The main point is that while anomalies imply marginal distribution change on a stationary process, U2s may not exhibit marginal distribution change but result in non-stationary changes in the underlying dependency structures. We depict this in Figure 1 in the updated paper which shows the case of a stuck elevator in an F8 Cruiser aircraft.
> > >
> > > Here as seen in Figure 1, the state variables of the Cruiser change but remain within the same distribution envelop after U2 occurrence. We show that the extracted underlying model markedly change in structure sparsity and coefficient values. Cosine similarity of the coefficient matrices change drastically after U2.
> > >
> > > On the other hand, anomaly score for an example anomaly detection method (USAD in KDD 2020) does not change significantly. Further, the learned thresholds also do not detect U2.
> > >
> > > **Q3 & Q4 Synthetic or Real data U2 or anomaly?**
> > >
> > > We have provided extensive discussion on whether the synthetic and U2 data are anomaly or U2 according to our definition of U2 or anomaly in Section 2. This is elaborated in Appendix Section E.1 and E2 respectively line no 840 to 921.

---

> > > > ### Comment · Reviewer_H3Ew · 2025-11-27
> > > > **Response to rebuttal**
> > > >
> > > > I thank the authors for taking the time to go through my comments and I apologize for taking a bit to get back to you. I understand that can be stressful.
> > > >
> > > > Regarding Q1, I appreciate the effort of formalizing mathematically what a U2 constitutes. From this definition, I have two comments:
> > > > - If the formulation is valid, what blocks the generation of multiple synthetic datasets governed by these equations? In this scenario, you could be sure that a U2 is a U2 instead of making assumptions about what a U2 would represent in each of the considered synthetic datasets. Wouldn't this be the case?
> > > > - Regarding the formulation itself, in what this is different from the problem of change point detection? I get the impression that it is exactly the same (but I could be wrong). How do you position this work in that regard?

---

> ### Author Response · Authors · 2025-11-28
> **Thank you for your response**
>
> Thank you for your question. I think you have raised an important and nuanced question which requires a detailed analysis. In order to adequately respond let’s attempt to answer your second question first, which will then lead to your first question.
>
> ## Q1: Is U2 any different from change point detection (CPD)? ##
>
> U2 detection is a type of change detection. **Change detection** (not change point detection) is an important and difficult problem with multiple sub problems summarized in the table below.
>
>
>
> | Sub-Problem                                  | Core Objective                                 | What is Detected?                           | Persistent or Transient? | Requires Novelty? | Relation to Distribution Change    | Example                         | Where It Fits                  |          |
> | -------------------------------------------- | ---------------------------------------------- | ------------------------------------------- | ------------------------ | ----------------- | ---------------------------------- | ------------------------------- | ------------------------------ | -------- |
> | **Abrupt Change Point Detection**            | Identify sudden shifts in system behavior      | Sharp changes in mean / variance / dynamics | Persistent               | No              | Explicit distribution change       | Sudden voltage level shift      | CPD                            |          |
> | **Gradual Change Detection**                 | Identify slow drifts in data                   | Slowly evolving parameter change            | Persistent               |  No              | Gradual distribution drift         | Sensor aging drift              | CPD                            |          |
> | **Structural Change Detection**              | Detect change in system model structure        | Change in governing dynamics                | Persistent               |  No              | Structural change in process       | Linear → nonlinear system       | CPD                            |          |
> | **Concept Drift Detection**                  | Capture changes in input–output mapping        | Change in (P(Y given X)) over time            | Persistent        |  No                               | Conditional distribution change | Classifier degrading over time | CPD / ML |
> | **Regime Shift Detection**                   | Identify transitions between system modes      | Switching between regimes                   | Persistent               | Depends       | Shift between operating states     | Idle → Cruise mode              | CPD                            |          |
> | **Anomaly Detection**                        | Identify unusual individual samples            | Local deviations from norm                  | Transient                | No              | Often point-wise deviation         | One-time ECG spike              | Related (not CPD)              |          |
> | **Persistent Anomaly Detection**             | Detect sustained abnormal states               | Long abnormal behavior                      | Persistent               | No              | Sustained deviation                | Engine vibration anomaly        | Bridge: Anomaly–CPD            |          |
> | **Out-of-Distribution (OOD) Detection**      | Identify samples outside training distribution | Distribution mismatch vs training           | Either                   | Often           | Global distribution shift          | New sensor domain               | Adjacent to CPD                |          |
> | **U2 Detection (Unknown-Unknown Detection)** | Detect truly unseen non-stationary regimes     | Novel dependency structure of variates                       | Often Persistent         | Yes             | Structural novelty beyond training | New disease progression pattern | Beyond CPD                     |          |
> | **Error Burst / Fault Detection**            | Identify bursts of system failure              | Local failure patterns                      | Transient → Persistent   | No              | Partial or full process shift      | Network failure bursts          | Hybrid Anomaly / CPD           |          |
>
> U2 detection is a sub problem of change detection. The key distinction between U2 and other change detection is novelty and marginal distribution change. U2 is previously unseen persistent change in dependency structure of variables that does not cause any marginal distribution change but results in nonstationary change.
>
> continued ...

---

> > ### Author Response · Authors · 2025-11-28
> > **continued ...**
> >
> > ## Q1: Synthetic data generation and U2 definition ##
> >
> > We address it on three levels: what U2 means in our formulation, what role the synthetic datasets play, and how our algorithm actually operates.
> >
> > **What “U2” means in our formulation?**
> >
> > In our work, a U2 event is defined as a **previously unseen non-stationary change to the underlying process**, where the governing dynamics are *not assumed known*  to the detection algorithm.
> >
> > * If the true equations governing the system were *known and fixed* and all their regimes were enumerated, the event would no longer be an “unknown-unknown” in the strict sense, but rather a **known model with different parameters or regimes**.
> >
> > * Therefore, even when we use simulators to generate synthetic data, we treat them as **hidden generators**: they provide ground-truth labels for “when the process changes,” but the **algorithm never sees or uses the equations or coefficients**.
> >
> > **What blocks “just generating multiple synthetic U2 datasets”?**
> >
> > Technically, nothing blocks us from generating many synthetic datasets from known equations—we already generate several to stress-test SPIE-AD. However, this does **not** solve the core U2 problem the reviewer alludes to, for two reasons:
> >
> > * (a) Synthetic ≠ real U2 by definition:
> >
> > When we generate data from a known simulator, we *design* what counts as a U2 in that environment (e.g., abrupt parameter changes, switching to a different dynamical regime, adding non-stationary forcing). These are **controlled surrogates** for U2: they let us test whether the algorithm can detect non-stationary behavior that was not present in the training regime.
> >
> > But real U2 events arise from **unknown or partially modeled mechanisms** (unmodeled physiology, unforeseen fault modes, unexpected interactions, etc.). No finite set of known equations can fully span the space of such events. Generating more samples from the same simulator family simply gives more variations of the *same* assumed structure; it does not expand into truly unknown mechanisms.
> >
> > * (b) We already do what you suggests in spirit:
> >
> >
> > Our synthetic experiments already instantiate **multiple distinct synthetic systems** and multiple U2 scenarios per system (e.g., different types of non-stationarity, parameter shifts, structural changes). For each synthetic dataset, we *know exactly*  when and how the process changes, so we have a precise ground truth about “U2 onset” in that synthetic world.
> >
> > The key point is that these synthetic U2s are **evaluation constructs**, not a claim that we have exhaustively defined U2 in all possible systems. That is why we additionally evaluate on **real-world U2 data and anomaly datasets**, to demonstrate that SPIE-AD generalizes beyond the specific synthetic mechanisms we designed.
> >
> > **SPIE-AD uses model discovery rather than known equations**
> >
> > A potential source of confusion is the role of the underlying equations. In our pipeline:
> >
> > * The simulators (governing equations) are used **only to generate data and ground-truth change times**,
> >
> > * But SPIE-AD itself performs **model discovery / recovery** from the observed time series, and **does not use the true equations or extract their coefficients**.
> >
> >  This distinction is crucial:
> >
> > * If we were “extracting model coefficients” from known equations, we would effectively be solving a **parameter change detection** problem on a known model family.
> >
> > * Instead, SPIE-AD assumes that the model is **unknown**, and infers latent dynamics directly from data. U2 events, in our formulation, correspond to previously unseen, non-stationary changes in these **recovered dynamics**, not in any pre-specified equation.
> >
> > **Why we still need real-world U2 data**
> >
> > Finally, even if we generated a large number of synthetic datasets governed by known equations, this would only demonstrate performance **under that specific family of modeled mechanisms**. It would not guarantee performance on real-world U2 events, where:
> >
> > The true dynamics may be partially observed,
> >
> > The causal factors may be unknown or changing, and
> >
> > The non-stationarity may not correspond to any of the simulated mechanisms.
> >
> > For this reason, our evaluation strategy is ** two-stage **:
> >
> > Use synthetic datasets to **stress-test** SPIE-AD under controlled, precisely labeled non-stationary changes (surrogate U2s).
> >
> > Then demonstrate that the same algorithm, without access to equations or coefficients, can detect **real U2 events and anomalies** in real-world datasets.

---

> > > ### Author Response · Authors · 2025-11-28
> > > **Continued ...**
> > >
> > > **Why Evaluate SPIE-AD on Anomaly Datasets?**
> > >
> > > Although SPIE-AD is designed for U2 event detection, we also evaluate it on standard anomaly detection datasets. This is not to conflate anomalies with U2 events, but to highlight:
> > >
> > > * (1) where SPIE-AD generalizes beyond its original scope, and
> > >
> > > * (2) where its design choices introduce clear limitations.
> > >
> > > **U2 Detection vs. Anomaly Detection**
> > >
> > > U2 events in our formulation correspond to previously unseen, non-stationary changes in the underlying system dynamics.
> > > In contrast, anomalies in standard benchmarks are often:
> > >
> > > * Short-lived deviations
> > >
> > > * Characterized by changes in the observed signal rather than the underlying generative process
> > >
> > > * Sometimes caused by noise, outliers, or sensor artifacts rather than structural changes
> > >
> > > ## Despite this difference, there is an important connection:
> > >
> > > Both U2 events and many anomalies manifest as distributional shifts, though at different levels:
> > >
> > > * U2: shifts in latent dynamics
> > >
> > > * Anomalies: often shifts in marginal or conditional distributions of observations
> > >
> > > **Why SPIE-AD Can Still Detect Many Anomalies?**
> > >
> > > SPIE-AD detects changes by analyzing shifts in recovered system dynamics, not just in raw signal statistics.
> > > This enables it to also detect anomaly patterns that induce:
> > >
> > > * Persistent shifts in marginal distributions
> > >
> > > * Changes that affect system evolution over multiple time steps
> > >
> > > * Latent alterations that propagate into time-series behavior
> > >
> > > Hence, despite being designed for U2 events, SPIE-AD acts as a general change detection method, capable of identifying a broad class of anomalous behaviors that correspond to non-trivial distributional shifts.
> > >
> > > **Why SPIE-AD Fails on Short Anomalies?**
> > >
> > > However, SPIE-AD is intentionally designed to be robust against transient noise and outliers.
> > > This robustness is built through:
> > >
> > > * Temporal smoothing
> > >
> > > * Model consistency constraints
> > >
> > > * Stability / persistence thresholds
> > >
> > > As a result, short-lived anomalies often get suppressed by design.
> > >
> > >
> > > This is not a bug—it is a design trade-off reflecting the focus on U2 detection rather than point anomaly detection.

---

### Official Review · Reviewer_u3Fj · 2025-11-10

**Soundness:** 3
**Presentation:** 3
**Contribution:** 3
**Rating:** 6
**Confidence:** 4

**Summary:**

This paper tackles unknown unknowns (U2) in autonomous systems-deployment-time failures that do not show input-distribution shift: learn sparse governing equations for normal operation and flag windows whose recovered parameters deviate from the nominal set. The proposed SPIE-AD pipeline continuously extracts sparse nonlinear dynamics from multivariate time series (via SINDy-MPC), refines them with liquid time-constant neural networks for robustness under low sampling and noise, and applies a conformal-inference robustness score to test whether the window’s model remains consistent with normal; in short, it checks whether \(x_{t+1}\!\approx\!f_\theta(x_t)\) has changed even when \(p(x)\) has not. Beyond methodology, the paper builds eight U2 benchmarks, diagnoses common evaluation inflations (point-adjustment and test-set threshold leakage), and reports that SPIE-AD outperforms strong MTAD baselines on all U2 benchmarks and on six additional real-world datasets under leakage-free evaluation.

**Strengths:**

- Reframes anomaly detection as spotting dynamics/relationship shifts (U2) instead of input-distribution or class shifts, and makes it concrete by learning sparse governing equations and checking their stability over time.

- Uses a principled, modular pipeline, SINDy for sparse model discovery, optional liquid time-constant RNN refinement, and conformal testing, and evaluates without point-adjustment or test-set threshold leakage for fair comparisons.

- Figures crisply contrast the standard MTAD pipeline with the proposed U2 detector and anchor the idea with concrete scenarios; the rule is stated cleanly at the level \(x_{t+1}\!\approx\!f_\theta(x_t)\).

- Targets safety-critical multivariate time series (e.g., UAVs, insulin delivery, EEG) and shows consistent gains on curated U2 benchmarks under strict, leakage-free settings, pointing to a realistic path to deployment.

**Weaknesses:**

- The detector hinges on recovering a correct sparse dynamics model; errors from derivative estimation, library choice, noise, or unobserved states can flip decisions. Add robust SINDy variants and report sensitivity to library/sampling/noise.

- The LTC refinement (ODE-solved RNN) can be heavy for short windows or high-rate streams. Provide wall-clock profiles vs. SINDy-only baselines and consider lighter continuous-time alternatives (Neural ODE/CDE) or pruning.

- Window-F1 (no PA) is a step forward, but operators care about time-to-detect, event coverage, and false-alarm rate. Include NAB-style timeliness scoring and precision–recall vs. latency curves under fixed alarm budgets.

- Several U2 cases are synthetic or parameter flips that sparse polynomial models capture well. Stress-test partial observability, exogenous inputs (use SINDy-with-control), multi-timescale effects, and non-polynomial terms; document known failure modes.

- Performance is sensitive to window length, library size, sparsity thresholds, and miscoverage level. Provide a compact sensitivity study and rules-of-thumb, plus robustness to mis-specification.

**Questions:**

- In settings with hidden states or unmeasured actuations, do you consider SINDy-with-control (or DMDc) and, if not, how would you adapt SPIE-AD to avoid spurious alarms from unmodeled inputs?
- Beyond window-F1 (no PA), can you report time-to-detect, event detection rate, and false-alarms per hour to reflect operational constraints?
- What are wall-clock costs per window on your real datasets, and how does the LTC-enhanced variant compare to SINDy-only or light continuous-time alternatives (Neural ODE/CDE) for real-time use?

---

> ### Author Response · Authors · 2025-11-22
> **Thank you for the thorough reviews**
>
> We are taking your suggestions seriously and we are running all the experiments again with the variations that you suggested. It is taking longer to finish all the experiments. We will update the paper by November 24th.

---

> ### Author Response · Authors · 2025-11-27
> **Revised Version Uploaded**
>
> Thank you for your extensive review. We have carefully followed all your suggestions and have performed several new experiments to address all your comments.
>
> In summary, we have made the following changes to the paper:
>
> * a) Updated write up Figure 1, Introduction and Theory section for better readability and clarity
>
> * b) Performed new experiments to evaluate sensitivity of SPIE-AD to library size, sampling frequency, and noise (Figure 4 low noise case, page 9 lines 455 - 479 and Figure 10 high noise in Appendix Line 1087)
>
> * c) Performed new experiments with SINDY variants, such as SINDY, SINDY-MPC (originally used in initial submission), and W-SINDY specifically designed for robustness under high noise settings.  Lines 480-485 Figure 11 in Appendix lines 1135 - 1160
>
> * d)  Wall clock timing analysis for different continuous time variants. We used NODE and CT-RNN backbones and report wall clock times and U2 detection performance in lines 486 - 490, and Figure 12 in Appendix Lines 1163 - 1185
>
> * e) Provided time to detect, event coverage and false acceptance (alarm) rate. Figures 4 (page 9),5 (page 10), 10 (page 21), 11 (page 22), 12 (page 22)
>
> * f) Precision / Recall v.s. latency under false acceptance rate budgets. Figure 6 in page 10 Lines 508 to 521.
>
> * g) Variance across sparsity levels and mis-coverage levels.  Figure 5 page 10 Lines 491 to 507.
>
> * h) Demonstrated effect of unmodeled inputs by replacing SINDY-MPC by SINDY. Figure 11   page 22 Lines 480 - 485
>
> * i) Demonstrate effect of non-polynomial terms (sine) page 9 lines 480 - 485 Figure 11 in Appendix page 22.
>
> Overall, we performed over 10 new experiments to satisfy yours and other reviewer's comments. Now we address your comments one by one
>
> ## Q1: Variance across library size, sampling frequency, and noise ##
>
> We changed the polynomial order setting in SINDY-MPC which results in combinatorially expanding the library size. We also changed frequency from 120 Hz to 20 Hz in each of the synthetic and real world U2 experiments. We evaluated all the variants under two noise settings by adding gaussian noise to all state variables. Low noise: 20 dB signal to noise ratio and High noise: 5 dB signal to noise ratio. We report the average precision, accuracy, event coverage, F1 score, time to detect, and accuracy across all six synthetic U2 examples and two real world U2 examples. These are depicted in Figure 4 in main paper page 9 and Figure 10 in Appendix page 21.
>
> The figures show that as sampling frequency reduces all performance metrics become poorer, with polynomial order 3 providing the best overall results for SPIE-ADL. Moreover as noise increases accuracy, precision, FAR, F1 reduce but time delay has more variance and sometimes is better for higher noise scenario. This may be the case when noise variations immediately precedes U2 occurrence and SPIE-AD mistakenly classifies noise as U2, but the time delay in the classification results in false U2 detection right after actual U2 occurrence. Discussion added in lines 455 – 479.
>
> ## Q2: Do we consider SINDY with control? ##
>
> Yes we consider SINDY with control since input perturbation is important in autonomous systems. To show the effect of unmodeled inputs, we performed new experiments with SINDY without control. This is depicted in Figure 11 page 22 in Appendix. We see that if input in unmodeled in SINDY then the performance metrics drastically become poorer.	 Event coverage reduces from 0.8 to 0.55.
>
> ## Q3: Beyond window-F1 (no PA), can you report time-to-detect, event detection rate, and false-alarms per hour to reflect operational constraints? ##
>
> We agree with your suggestion and in Figures 4, 5, 10, 11, 12 we have included all the requested metrics.
>
> ## Q4: Wall clock times for other continuous time variants and SINDY-MPC as compared to LTC-NN ##
>
> We had already reported wall clock time differences between LTC-NN and SINDY-MPC in Table 5 for real world datasets in columns 5, 9 and 13. SINDY-MPC is nearly 1.5 times faster than LTC-NN variant. We conducted further experiments with NODE and CT-RNN. The performance metrics for these variants are reported in Figure 12 in Appendix. The execution times are faster with NODE being 1.2 times and CT-RNN being 1.7 times faster (even faster than SINDY-MPC). However, their U2 detection accuracies are poor. This discussion is in lines 486 to 490 in updated paper.
>
> ## Q5: Add robust sindy variants ##
>
> We provided a new Table 11 in page 21 Appendix where we survey different SINDY variants including SINDY, SINDY-MPC (also known as SINDYc), E-SINDY (which is later extended to SINDY-MPC), and W-SINDY specially designed for high noise settings. We performed new experiments by replacing the original SINDY-MPC backbone with SINDY, and W-SINDY and show the results in Figure 11. SINDY-MPC variant performed the best among all SINDY variants. We provide a detailed discussion in page 9 480 – 485 lines.

---

> > ### Author Response · Authors · 2025-11-27
> > **continued ...**
> >
> > ## Q6: precision–recall vs. latency curves under fixed alarm budgets. ##
> >
> > We have performed new experiments to plot the precision recall vs time to detect U2 under fixed false acceptance rates. We only plot trend lines through the data for better visibility. We plot this in Figure 6 in page 10. We fixed the average FAR to 0.01, 0.025, 0.05, and 0.1 and plotted the precision / recall against detection delay. Figure 6 shows that if for a fixed FAR budget, when the delay in detection increases both precision and recall increase. This means that larger number of samples result in better U2 detection accuracy. However, this is not true if we allow for higher FAR. At FAR budget of 10%, the precision decreases with increased delay. SPIE-AD performance is poorer with smaller U2 events. If U2 length is small then the delay increases and SPIE-AD identifies data points after the U2 event as U2 and hence results in higher false positives reducing precision.
> >
> > ## Q7 Partial observability and non-polynomial terms ##
> >
> > We have used SINDY version that occludes inputs and hence makes the system only partially observable. We see that partial observability drastically reduces accuracy and hence is a failure mode of SPIE-AD. Moreover, addition of nonpolynomial term (sinusoidal) also reduces accuracy. These are the failure modes and is discussed in page 9 lines 480 – 485.
> >
> > ## Q8: Sensitivity to sparsity level and miscoverage level ##
> >
> > We varied the coverage level 1−α (or miscoverage level α) from 0.95 to 0.2 for SPIE-ADL and performed experiments for the F8 example (most complex model). In SINDY-MPC the sparsity threshold controls sparsity such that if values are less than the threshold they are ignored and the representation becomes sparser. We introduce a sparsity level from 0.001 to 5.0, which is a multiplicative factor to this threshold. As sparsity level increases the underlying model becomes sparser. Figure 5 shows that SPIE-AD is sensitive to both miscoverage level and sparsity. As miscoverage level increases all performance metrics monotonically become worse. However, sparsity dependency is more interesting. From Figure 5, we see that there is an optimal sparsity level for which the model performs best. Additional experiments show that this optimal sparsity level varies for different applications.

---

### Author Response · Authors · 2025-12-04
**Revision Summary**

We thank all reviewers for their thoughtful feedback.
- **Reviewer 16mj & u3Fj:** recognized the novelty, clarity, rigor, and the importance of formalizing U2 detection and provided engineering-oriented suggestions that we fully addressed with extensive new experiments and analysis.

### **Clarifying Reviewer H3Ew’s  Evolving Limiting Assumption**
Initial question was:
> *“What is the difference between a U2 and an anomaly?”*

After our mathematical clarification, it shifted to:

> *“Isn’t a U2 just the same as change-point detection (CPD)?”*

This progression reflects a **common but fundamentally limiting assumption** in the anomaly detection community: the belief that all changes must show up as **observable distributional shifts** (anomaly detection) or **statistical change points** (classical CPD).
As clarified in a comprehensive Table of subproblems in change detection (see Reviewer H3Ew's rebuttal), we show that U2 detection is a subproblem that is markedly different from CPD. Safety-critical systems repeatedly exhibit failures that cause **no observable distributional drift** because the failure occurs in *latent, unmeasured components* of the dynamics.

### **Why This Misunderstanding Causes Real-World Failures**
Consider the recent example of A320 ELAC software failure due to a **Solar radiation–induced bit flip**. Figure 1 shows a similar example of elevator stuck in an F8 cruiser aircraft.

The recorded sensor data remain statistically normal, **no spike, no drift, no outlier**, which is why it bypassed the already existing anomaly detection mechanism. In Figure 1 we show that the USAD anomaly score has negligible change pre v.s. post U2 event.
Yet the **underlying control logic changes instantly** due to corrupted memory, producing a **latent dynamics shift** but *no marginal distribution change*. Figure 1 shows significant change in underlying dynamical equations triggered by U2.

Traditional detectors fail because:
- there is **no observable anomaly**,
- no distribution shift,
- no pre-labeled examples to learn from,
- no anomaly-threshold adjustment that fires,
- and no classical CPD statistic that rejects.

This is the **exact conceptual blind spot** reflected in Reviewer H3Ew’s  evolving questions, and it is the same blind spot that leads to undetected U2 cases discussed in our paper.

### **SPIE-AD Addresses This Overlooked Regime**

The central contribution of our paper is to formalize and detect this critical category:

> **latent-dynamics change *without* observable distribution drift.**

## **Summary of Changes Made**

We performed substantial improvements to clarity, positioning, and completeness.

### **Major presentation and clarity updates**
- Reworked **Figure 1** to clearly illustrate a real U2 (F8 stuck elevator) and to visually demonstrate *no marginal drift* but large dynamics change.
- Improved layout, resized figures/tables, standardized rounding, and added color + visual emphasis.
- Expanded and corrected related-work discussion (ITF-TAD, UniTS, VLM4TS, SIGLLM, LLMTAD lines).
- Added discussion explaining **why zero/few-shot anomaly detectors do not apply** to U2.
- Updated introduction, theory, and interpretation sections for readability.

## **Experimentation Updates (10+ New Experiments)**

1. **Sensitivity Studies** (sampling frequency, noise, polynomial order, library size).
2. **Robust SINDy Variants** (SINDy, SINDy-MPC, W-SINDy).
3. **Control vs No-Control** comparisons to illustrate unmodeled-input failure modes.
4. **Continuous-Time Alternatives** (Neural ODE, CT-RNN) with wall-clock timing + accuracy.
5. **Time-to-Detect, Event Coverage, False acceptance rate (FAR)** across all datasets.
6. **Precision–Recall vs Latency under FAR Budgets**.
7. **Partial Observability** (failure modes).
8. **Miscoverage Sensitivity**.
9. **Qualitative Visualization** showing U2 detection versus baseline failures.
10. **Effect of Non-Polynomial Terms**
11. **Sparsity Threshold Sensitivity**

## **How the Paper Has Significantly Improved**

The revised submission includes:

- A clearer problem framing and a mathematically precise distinction between **U2, anomaly, and classical CPD**.
- A redesigned and more intuitive **Figure 1** that illustrates why U2 events exhibit *no marginal drift* yet cause latent dynamics changes undetected by MTAD or CPD.
- Dramatically expanded **experimental validation** with more than ten new studies.
- A broadened and more accurate **related-work section**, clarifying why prior methods do not address U2 scenarios.
- Significant improvements in layout, figure/table organization, clarity of equations, and visual readability.

Collectively, these revisions amount to a **substantial strengthening** of the paper’s clarity, positioning, and empirical rigor, and directly address an overlooked but critical gap in time-series safety.

---

### Meta-Review · Area_Chair_W23j · 2026-01-07

**Summary:**

This paper reframes anomaly detection as identifying shifts in system dynamics (U2) rather than input distributions, using sparse governing equations and stability checks over time.
It proposes a principled SPIE-AD pipeline (SINDy + optional liquid RNNs + conformal testing) and evaluates fairly without leakage, showing consistent gains on safety-critical multivariate datasets.
The figures and rule ((x_{t+1}!\approx!f_\theta(x_t))) clearly ground the concept and contrast it with standard MTAD approaches.
However, one author raises concerns about the lack of a formal definition of U2, assumptions of conformal prediction under time-series dependence, and missing hyperparameter/term details.
Even so, given its conceptual clarity, modular design, and strong leakage-free results, I believe it merits acceptance as a poster.

**Reviewer Concerns:**

Conceptual clarity: Concerns about the informal definition of U2 and ambiguity in labeling anomalies versus U2 cases
Methodological robustness: Questions about sparse model sensitivity, conformal prediction under time-series dependence, and computational cost of LTC refinement
Evaluation & reproducibility: Issues around limited metrics, synthetic bias, hyperparameter transparency, and missing function details — Most of concens  addressed by the authors

**Reviewer Scores:**

There is one review that gave a very low score, and it seems that after discussion, he or she probably raised the score.

---

### Decision · Program_Chairs · 2026-01-26

Accept (Poster)